# Interventional hydrogel microsphere vaccine as an immune amplifier for activated antitumour immunity after ablation therapy

Xiaoyu Liu [1,5], Yaping Zhuang [2,5], Wei Huang[1,5], Zhuozhuo Wu [1], Yingjie Chen[1], Qungang Shan[1], Yuefang Zhang[3], Zhiyuan Wu[1], Xiaoyi Ding[1], Zilong Qiu [3], Wenguo Cui [2,6] ✉ & Zhongmin Wang [1,4,6] ✉

The response rate of pancreatic cancer to chemotherapy or immunotherapy pancreatic cancer is low. Although minimally invasive irreversible electroporation (IRE) ablation is a promising option for irresectable pancreatic cancers, the immunosuppressive tumour microenvironment that characterizes this tumour type enables tumour recurrence. Thus, strengthening endogenous adaptive antitumour immunity is critical for improving the outcome of ablation therapy and post-ablation immune therapy. Here we present a hydrogel microsphere vaccine that amplifies post-ablation anti-cancer immune response via releasing its cargo of FLT3L and CD40L at the relatively lower pH of the tumour bed. The vaccine facilitates migration of the tumour-resident type 1 conventional dendritic cells (cDC1) to the tumour-draining lymph nodes (TdLN), thus initiating the cDC1-mediated antigen cross-presentation cascade, resulting in enhanced endogenous CD8$^+$ T cell response. We show in an orthotopic pancreatic cancer model in male mice that the hydrogel microsphere vaccine transforms the immunologically cold tumour microenvironment into hot in a safe and efficient manner, thus significantly increasing survival and inhibiting the growth of distant metastases.

Pancreatic ductal adenocarcinoma (PDAC) is a lethal digestive tract cancer with a 5-year survival rate of less than 7%[1]. More than 70% of patients have missed the opportunity for surgery when they are diagnosed with pancreatic cancer[2,3]. Current chemotherapy regimens for pancreatic cancer are limited with systemic toxic effects. Thus, minimally invasive intervention therapy is a viable option for unresectable pancreatic cancers. Irreversible electroporation (IRE) is an emerging nonthermal ablation therapy[4–6]. It takes advantage of high-frequency electric fields to induce nanoscale perforation in cell membranes, perturbing cell homoeostasis and resulting in strong immunogenic cell death (ICD) and antigen release[4,7]. IRE ablation destroys tumours without damaging the surrounding blood vessels and nerves and effectively reduces the risk of pancreatic fistula formation and bleeding, showing advantages in the treatment of localised solid tumours[6,7]. However, patients receiving IRE treatment still suffer from rapid recurrence and metastasis of pancreatic cancer[7,8]. This recurrence is due to the typical dominant immunosuppressive tumour microenvironment (TME) of the

[1]Department of Radiology, Ruijin Hospital, Shanghai Jiao Tong University School of Medicine, No.197, Ruijin 2nd Road, 200025 Shanghai, P. R. China. [2]Department of Orthopaedics, Shanghai Key Laboratory for Prevention and Treatment of Bone and Joint Diseases, Shanghai Institute of Traumatology and Orthopaedics, Ruijin Hospital, Shanghai Jiao Tong University School of Medicine, 197 Ruijin 2nd Road, 200025 Shanghai, P. R. China. [3]Institute of Neuroscience, CAS Key Laboratory of Primate Neurobiology, State Key Laboratory of Neuroscience, CAS Center for Excellence in Brain Science and Intelligence Technology, No.320 Yueyang Road, 200032 Shanghai, P. R. China. [4]Department of Radiology, Ruijin Hospital Luwan Branch, Shanghai Jiao Tong University School of Medicine, No.149, South Chongqing Road, 200025 Shanghai, P. R. China. [5]These authors contributed equally: Xiaoyu Liu, Yaping Zhuang, Wei Huang. [6]These authors jointly supervised this work: Wenguo Cui, Zhongmin Wang. ✉e-mail: wgcui@sjtu.edu.cn; wzm11896@rjh.com.cn

disease, including insufficient immune cell infiltration, impaired immune cell function, and immunosuppressive molecule accumulation[9–11]. The cold TME promotes immune evasion and rapid tumour recurrence after ablation therapy[8,12]. Therefore, immune reactivation is the key to improving the antitumour therapeutic effect after tumour ablation.

In clinical practice, immune checkpoint blockade (ICB) is the most commonly utilised immunotherapy. However, blocking PD-1/PD-L1 or combining it with CTLA-4 inhibition has been found to be ineffective except in extremely rare microsatellite-unstable cases of pancreatic cancer[1,13,14]. Recent studies demonstrated that PD-1/PD-L1 inhibition could therapeutically synergise with IRE ablation in preclinical cancer models. This synergistic effect is mostly because of increased tumour-associated antigen (TAA) release and enhanced immune infiltration[4,15,16]. However, clinical evidence indicates that patients who receive combination therapy for pancreatic cancer continue to face an increased risk of tumour recurrence[7,17]. In fact, the efficiency of immunotherapy is highly dependent on both immune infiltration and immune activation. Although ICB prevents T-cell exhaustion, they have little effect on the amplification of endogenous antitumour immunity[18]. Therefore, triggering efficient amplification of endogenous adaptive immune responses is critical for improving immunotherapy after ablation therapy[19].

The antigen cross-presentation process is critical to the antitumour activity of endogenous T cells[20]. Recent advances in nanoparticle-based immunisations have focused on the codelivery of exogenous antigens or adjuvants to antigen-presenting cells (APC) to amplify tumour-specific T cells[21–23]. In this process, antigen cross-presentation by tumour-resident type-I-conventional dendritic cells (cDC1) via major histocompatibility-complex class I (MHC-I) is necessary for a robust antitumour response of CD8+ T cells[24–26]. However, compared with endogenous TAAs, the exogenous antigen- or adjuvant-induced immune response cannot overcome the heterogeneity of tumours[27–29]. Hence, Liu et al. employed fluorinated dendritic macromolecules and polyethylene imine as antigen delivery vehicles to create individualised vaccines using antigens derived from postoperative tumour tissues, realising personalised postoperative immunotherapy[30]. However, antigen cross-presentation is a complex immune cascade, and each step of the cascade is critical for triggering adequate amplification of antitumour immunity[22,23,31]. Targeting the dynamic antitumour immune cascade using nanomedicine for cancer immunotherapy has not been deeply investigated[32,33].

CD103+ CD11b- cDC1s directly activate endogenous CD8+ T cells through a multistep cascade including intratumour recruitment, activation and antigen uptake, and migration to tumour-draining lymph nodes (TdLN) for CD8+ T cells activation[24–26]. Moreover, the deficiency

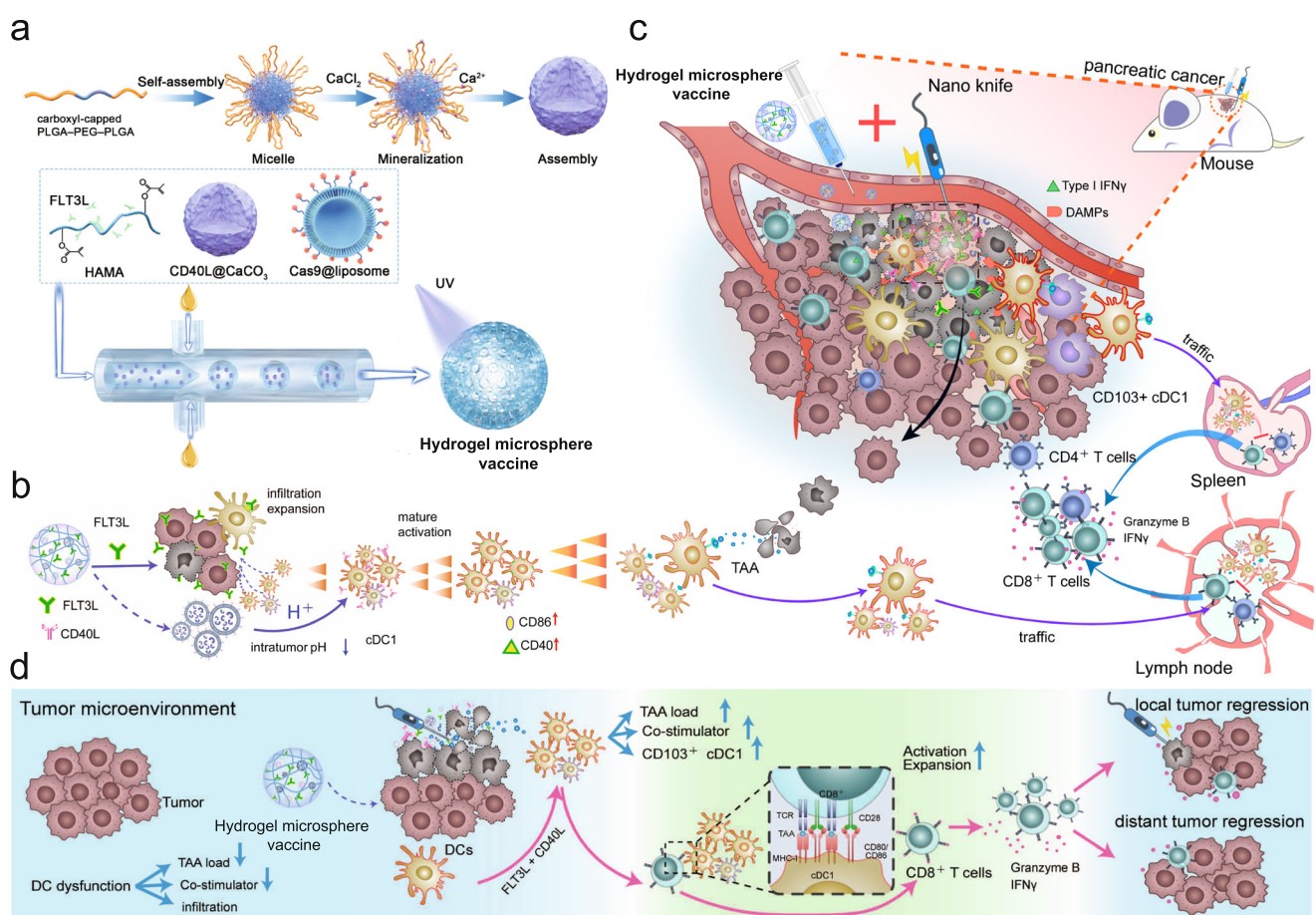

**Fig. 1 | The cDC1-activated hydrogel microsphere vaccine was used as a general immune amplifier to amplify the cDC1/CD8+ T-cell antitumour axis after ablation therapy. a** Schematic diagram of the generation of the hydrogel microsphere vaccine. Briefly, we loaded a Cas9 plasmid and CD40L cytokines into liposomes and CaCO₃ nanoparticles, respectively, and then mixed them with FLT3L cytokines and hyaluronic acid (HA) to create a hydrogel microsphere vaccine under microfluidic control. **b** The hydrogel microsphere vaccine amplified the recruitment and amplification of tumour-resident cDC1s by rapidly releasing FLT3L, followed by the controlled release of CD40L in the acidic TME, which further amplified cDC1 maturation and migration into TdLNs. **c** The hydrogel microsphere vaccine acts as an immune amplifier to trigger a rocket-like amplification of the CD103+CD11b- cDC1-mediated antigen cross-presentation cascade, resulting in dramatic amplification of the antitumour immunity of endogenous CD8+ T cells after ablation therapy. **d** The hydrogel microsphere vaccine induced strong systemic antitumour immunity, which combated the growth of distant metastases. Source data are provided as a Source data file.

of CD8[+] T cells caused by CD103[+]CD11b[-] cDC1 dysfunction is an immunosuppressive feature of a variety of solid tumours[34-36]. Thus, triggering the antigen cross-presentation cascade of CD103[+] CD11b[-] cDC1s is an ideal way to amplify CD8[+] T-cell antitumour immunity in pancreatic cancer. Recently, sequential intravenous administration of FLT3L or CD40 agonist has been shown to activate tumour-resident cDC1s in tumour-bearing mice[37,38]. However, this therapy is highly dependent on treatment-induced ICD and has a minimal remodelling effect on the TME. In addition, intravenous delivery of cytokines has the defect of a short drug half-life and side effects, including lethal cytokine storms and systemic immune disorders[39-41]. Biomaterials hold tremendous promise for improving the efficiency of cancer immunotherapies, and the hydrogel is an attractive candidate for local drug delivery systems that can deliver drugs to targeted sites[32,42]. Rationally designed hydrogel microspheres loaded with functional nanocarriers can improve drug access to tumour sites, directly reverse the immunosuppressive status of solid tumours, trigger specific tumouricidal immune effects, and prevent the formation of distant metastases, with minimal side effects[43-45].

Here, we develop an injectable hydrogel vaccine that stimulates the cDC1-mediated antigen cross-presentation cascade. In mice with pancreatic tumours, the hydrogel vaccine combined with ablation therapy effectively promotes systematic amplification of cDC1/CD8[+] T-cell antitumor immunity, inhibiting pancreatic cancer progression. This combination of immune amplification and ablation therapy presents a novel strategy for treating locally advanced pancreatic cancer. Our work holds significant relevance for advancing the field of cancer immunotherapy and improving outcomes for patients with this challenging disease.

## Results

### Rational design and function of the injectable hydrogel microsphere vaccine

Here, we designed a CD103[+] CD11b[-] cDC1-activated hydrogel microsphere vaccine as an "immune amplifier" to amplify the cDC1/CD8[+] T-cell antitumour axis after ablation therapy. Briefly, we loaded the Cas9 plasmid and CD40L into liposomes and CaCO$_3$ nanoparticles, respectively, and then mixed them with FLT3L cytokines and hyaluronic acid (HA) to create a hydrogel microsphere vaccine under microfluidic control (Fig. 1a). The hydrogel microsphere vaccine induces a rocket-like cascade amplification of cDC1-mediated antigen cross-presentation and CD8[+] T-cell activation, promoting the transformation of pancreatic cancer from "cold" to "hot" tumours in a safe and efficient manner (Fig. 1b, c). Finally, the hydrogel vaccine effectively activated systemic antitumour immunity, inhibited distant metastasis and improved survival in tumour-bearing mice (Fig. 1d).

### Preparation and functional validation of a hydrogel microsphere vaccine

We anticipated that liposomes would improve Cas9 plasmid transfection into cells in the electroporated region (Fig. 2a, b). We determined that the liposome packaging efficiency could reach 82.7% when the ratio of plasmid DNA to liposomes was 1:1 and that increasing the liposome ratio produced no significant improvement in the packaging efficiency (Fig. 2c). The liposomal nanoparticles had a diameter of -155 nm and a PDI (polydispersity index) value of 0.177 (Fig. 2d), indicating a relatively uniform size distribution. Next, we generated CD40L@CaCO$_3$ nanoparticles via coprecipitation (Fig. 2e, f) and assessed the size distribution of the CaCO$_3$ nanoparticles using DLS (Fig. 2g). We anticipated that the CaCO$_3$ nanoparticles could carry CD40L and programmatically release the drugs in the acidic TME. The terminal carboxylation of the tri-block polymer PLGA−PEG−PLGA was modified using succinic anhydride, with the signal peak in the nuclear magnetic spectrum ([1]H NMR) at a chemical shift of 2.68 ppm attributable to methylene protons (−CH$_2$CH$_2$−) of the succinyl unit (Fig. 1a and

Supplementary Fig. 1a). In the hydrogel system, carboxylated PLGA−PEG−PLGA (HOOC−PLGA−PEG−PLGA−COOH) self-assembled into homogeneous micelles with a particle size of approximately 32 nm (Fig. 2h). The −COOH groups of micelles further promoted mineralisation of Ca[2+] and the stabilisation of CaCO$_3$ nanoparticles. HA was chemically modified with methacrylate anhydride. [1]H NMR data indicated that the signal peaks at 5.66 and 6.10 ppm were attributable to methacrylate anhydride's double-bond hydrogen, with a double-bond modification rate of 69.8% (Supplementary Fig. 1b). Finally, we mixed the FLT3L peptide, Cas9@lip, and CD40L@CaCO$_3$ nanoparticles into HAMA and used microfluidics to generate the hydrogel microsphere vaccine (Fig. 2i, j and Supplementary Fig. 1c−h). As determined by energy dispersive spectroscopy (EDS), the CaCO$_3$ nanoparticles were uniformly dispersed in the hydrogel microsphere vaccine (Fig. 2k). Overall, we developed a hydrogel microsphere vaccine with an integrated genome editor and multiple bioresponsive nanomaterials for modulation of the dynamic immune response.

Then, we evaluated the efficacy of the hydrogel microsphere vaccine in releasing FLT3L and CD40L cytokines at pH=6.8, which is within the pH range in the majority of solid tumours. The cumulative release rate of FLT3L reached 48.5% at 24 h. By comparison, the release rate of CD40L was only 9.3% at 24 h but increased to 54.1% at 72 h, indicating that the hydrogel microsphere vaccine achieved the release of dual cytokines in response to an acidic environment (Fig. 2l). The pH dependence of CD40L release (Supplementary Fig. 2a) and the concentration of Cas9 plasmid in the hydrogel system (Supplementary Fig. 2b) were also further confirmed. We next investigated the effect of the combination of the hydrogel microsphere vaccine and electroporation on plasmid transfection into several cell lines (Fig. 2m). Flow cytometry analysis showed that the hydrogel microsphere vaccine transfected more than 60% of cells (Panc02, KPC, AsPC1 or 3T3 cells) after electroporation, almost twofold (P < 0.0001) higher than the rate achieved with in the bare plasmid electroporated groups (Fig. 2n, o). These results indicate that the hydrogel microsphere vaccine significantly increases the electroporation-mediated plasmid transfection of cells. Finally, we assessed the effect of the hydrogel microsphere vaccine on the maturation and activation of bone marrow-derived dendritic cells (BMDCs) in vitro (Fig. 2p and Supplementary Fig. 3). After preincubation with IL-4 and GS-CSF, the additional FLT3L or hydrogel microsphere vaccine significantly promoted the differentiation. of BMDCs into CD103[+] CD11b[-] cDC1-like cells (Supplementary Fig. 4a). The hydrogel microsphere vaccine also increased the proportion of Ki67[+]cDC1-like cells (Supplementary Fig. 4a), suggesting an improvement of the proliferation ability of the cDC1s. The hydrogel microsphere vaccine nearly doubled the expression of co-stimulatory molecules (CD80, CD86, CD40 and MHC-I) in CD103[+] CD11b[-] cDC1-like cells compared to that in blank microgel control cells (P < 0.0001) (Fig. 2q). In addition, the hydrogel microsphere vaccine had a minimal effect on the differentiation of M1-like macrophages, and slightly reduced the proportion of CD206[+] immunosuppressive M2-like macrophages in vitro (Supplementary Fig. 4b, c). Taken together, these findings indicated that the hydrogel microsphere vaccine achieved multiple designed roles, including the efficient delivery of genomic editors and activation of CD103[+] CD11b[-] cDC1s by controlled cytokine release.

### Irreversible electroporation ablation induces immune remodelling in pancreatic cancer

We first reconstructed the immune landscape of human pancreatic cancer using the RNA-sequencing (RNA-seq) data of the TCGA-PAAD cohort (n = 177) (Fig. 3a). The infiltration score of tumour-resident DCs was highly correlated with T-cell infiltration, CD8[+] T-cell infiltration and cytotoxic T-cell infiltration in human pancreatic cancers (Fig. 3b). Further analysis showed that the intratumour level of *FLT3LG* and *CD40LG* were positively correlated with αDC and tumour-resident DC

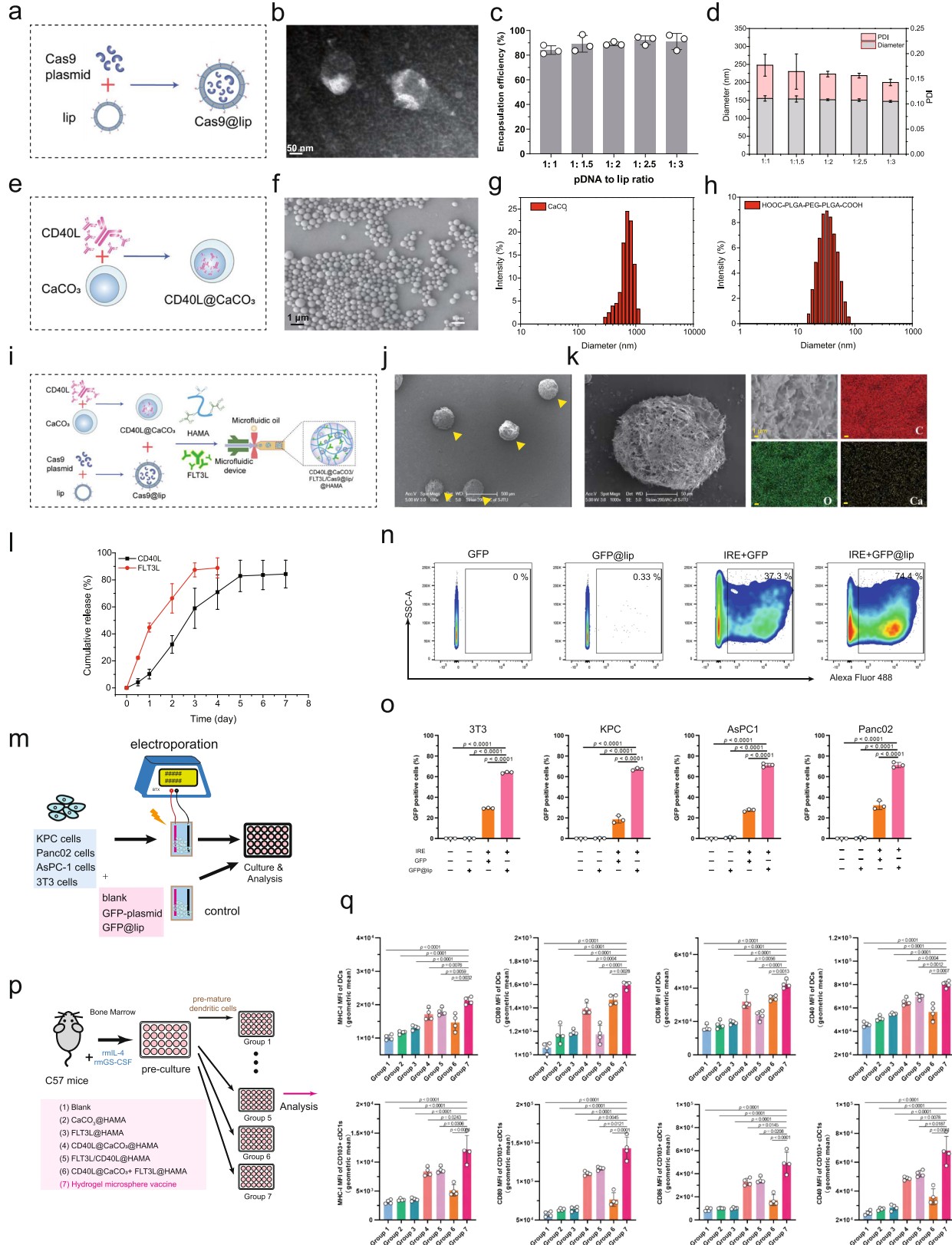

scores in human pancreatic cancers (Fig. 3c). Then, we established an immunocompetent orthotopic pancreatic cancer mouse model, and confirmed that the orthotopic pancreatic tumour had highly consistent pathological and immune features with the pancreatic tumours derived from $Kras^{LSL-G12D}$; $Tp53^{fl/-}$; $Pdx1$-Cre mice (Supplementary Fig. 5a, b). We explored the effect of IRE ablation on the TME using mice

bearing orthotopic pancreatic cancers (Fig. 3d). Significant tumour necrosis and collagenous fibres were identified in the ablation area after 7 days of IRE treatment (Fig. 3e, f). The positive TUNEL staining suggested cell apoptosis and potential debris release in the tumour region (Fig. 3e). The pH values were measured before ablation, 0, 24, 48 and 72 h after the ablation therapy. The mean pH value for each

**Fig. 2 | Preparation and functional validation of the hydrogel microsphere vaccine in vitro. a** Schematic diagram of the preparation of Cas9@liposome nanoparticles. **b** Representative TEM images of the plasmid@lip nanoparticles. **c, d** Encapsulation efficiency (**c**), particle size distribution and PDI value (**d**) of the Cas9@lip nanoparticles. $n = 3$ independent experiments. Data are presented as mean values +/− SD. **e** Schematic diagram of the preparation of the CD40L@CaCO$_3$ nanoparticles. **f** Representative scanning electron microscope (SEM) images of the CD40L@CaCO$_3$ nanoparticles. **g** Identification of the hydrodynamic diameter of the CD40L@CaCO$_3$ nanoparticles using the Malvern−Zetasizer method. **h** Identification of the hydrodynamic diameter of the HOOC−PLGA−PEG−PLGA−COOH solution. **i** Schematic diagram of the preparation of the hydrogel microsphere vaccine. **j** Representative scanning electron microscope (SEM) images of the hydrogel microspheres. **k** Enlarged representative SEM image (left panel) and regional element analysis (right panel) of the hydrogel microsphere vaccine. **l** The release of FLT3L and CD40L by the hydrogel microsphere vaccine in vitro (pH = 6.8). $n = 3$ independent experiments. Data are presented as mean values +/− SD. **m** Schematic diagram of cell transfection analysis using the combination of the hydrogel microsphere vaccine and electroporation. **n** Representative images of flow cytometry experiments identifying GFP-positive KPC cells in different treatment groups (**o**). **o** Analysis of the transfection efficiency of 3T3\KPC\AsPC1\Panc02 cells in different treatment groups. $n = 3$ biologically independent samples. **p** Schematic diagram illustrating the effect of hydrogel microsphere vaccines and different hydrogel counterparts on bone marrow-derived dendritic cells (BMDCs) in vitro. **q** Analysis of the effect of hydrogel microsphere vaccines and hydrogel microsphere counterparts on the expression of MHC-I, CD40, CD80 and CD86 on the surface of total DCs (CD24$^+$CD64$^-$CD11c$^+$MHCII$^+$CD45$^+$alive) and cDC1s (CD103$^+$CD11b$^-$CD24$^+$CD64$^-$CD11c$^+$MHCII$^+$CD45$^+$alive) in different treatment groups. $n = 4$ biologically independent samples. **b, f, j, k** Three independent experiments were performed to confirm data stability. **o, q** Data are presented as mean values +/− SEM. Two-tailed unpaired $t$ test. Source data are provided as a Source data file.

group is 6.84, 6.81, 6.75, 6.70, and 6.67, respectively (Supplementary Fig. 6). Our RNA-seq data revealed that IRE ablation significantly altered the pancreatic cancer transcriptome profile, with the expression of 2005 genes upregulated and that of 1446 genes downregulated (Fig. 3g). Gene set enrichment analysis (GSEA) revealed that inflammatory responses, adaptive immunity, IFN responses, and antigen-binding responses were significantly enhanced after ablation therapy (Fig. 3h, i). The levels of multiple immunoregulatory genes, including *CD274, Trbc2, Ifng, Gzmb, Flt3lg* and *CD40lg* were dramatically elevated (Fig. 3j). Immunohistochemical (IHC) staining showed that the ablation-induced proinflammatory environment promoted the infiltration of CD45$^+$ immune cells into tumour regions (Fig. 3k, l).

It's generally believed that IFN signalling acts as a double-edged sword for tumour immune modulation[46–48]. Decreased IFN signalling in pancreatic tumours can impair antigen presentation, hence impairing cytotoxic CD8$^+$ T-cell activation. However, overactivated IFN signalling after ablation therapy may also result in increased multigene-mediated immunotherapy resistance and Treg activation, which is a critical challenge in tumour immunotherapy. Multigene-mediated immunotherapy resistance is commonly acquired by increased expression of immune checkpoint ligands or receptors. Here, we designed a small Cas9 plasmid library targeting CD274 and used it to synthesise Cas9@lip nanoparticles for loading into the hydrogel microsphere vaccine (Supplementary Fig. 7). In combination with IRE ablation, local injection of hydrogel microspheres significantly attenuated the progression of orthotopic KPC pancreatic tumours in mice (Fig. 3m). The survival time of bearing orthotopic pancreatic tumour also significantly extended after the combination therapy (Fig. 3n). We used immunofluorescence staining and Western blotting to confirm the level of TNF-α, IFN-γ, PD-L1, HMGB1 and extrinsic SpCas9 across mice receiving different therapeutic combinations (Fig. 3o, p). The upregulation of HMGB1 is a critical molecular event of DAMP (danger-associated molecular pattern) signalling, implying the activation of the proinflammatory response induced by tumour necrotic debris after IRE ablation. The level of TNF-α, IFN-γ, PD-L1, HMGB1 and extrinsic SpCas9 was also significantly increased after IRE ablation (Supplementary Fig. 8a–d). Notably, the hydrogel microspheres effectively reduced PD-L1 expression after IRE ablation (Fig. 3o, p and Supplementary Fig. 8e). These findings suggest that the hydrogel microspheres act synergistically with IRE ablation to enhance the proinflammatory immune microenvironment, and suppress the expression of immune checkpoint ligand PD-L1 to facilitate cancer immunotherapy.

### The hydrogel microsphere vaccine amplified the recruitment, activation and migration of cDC1s

Most previous studies have focused on the delivery of exogenous antigens or immunoadjuvants to activate DCs. In comparison, we anticipated that the controlled cytokine release of dual cytokines could precisely amplify the recruitment, expansion, and activation of endogenous CD103$^+$ CD11b$^-$ cDC1s to achieve an adequate endogenous antitumour immune response in vivo (Fig. 4a). Meanwhile, we also prepared four kinds of control hydrogel microspheres. Group1 to Group4 represent CaCO$_3$/Cas9@Lip/@HAMA, FLT3L/Cas9@Lip/@HAMA, CD40L/Cas9@Lip/@HAMA and CD40L/FLT3L/Cas9@Lip/@HAMA, respectively (Fig. 4a). The mice receiving intratumour injection of hydrogel microsphere vaccine had increased tumour necrosis and immune cell infiltration comparing with the Group1 to Group4 counterparts (Fig. 4b and Supplementary Fig. 9a). In contrast with direct intratumour injection of the cytokines, delivery of the hydrogel microsphere vaccine achieved sequential intratumour release of FLT3L and CD40L. The highest intratumour concentration of FLT3L was detected at 24 h after injection, while the highest intratumour concentration of CD40L lasted from 48 to 96 h (Fig. 4c). Notably, Flow cytometry analysis showed that the hydrogel microsphere vaccine significantly amplified the proportion of CD103$^+$ CD11b$^-$ cDC1s among total tumour-resident DCs, increasing the proportion of cDC1s from 8.8 to 24.9% (Fig. 4d, e and Supplementary Figs. 9b, c and 10a). The hydrogel microsphere vaccine also significantly amplified the intratumour density of MHC II$^+$ CD11c$^+$ cells, which was ~1.5-, 1.7-, 2.2- and 2.0-fold greater than that of the Group1, Group2, Group3, and Group4 control hydrogel microspheres, respectively (Fig. 4e). The expression of CD86 and CD40 on the surface of cDC1s was substantially increased after hydrogel microsphere vaccine injection, with CD40 levels nearly 2.0-fold higher than those in Group1 controls (Fig. 4f), whereas the levels of PD-L1, the predominant inhibitory costimulator on DCs, kept on low after the vaccine stimulation (Supplementary Fig. 10b). Meanwhile, the expression of CD40 and CD86 on dnDCs and cDC2s were not significantly altered (Supplementary Fig. 10c, d). Further analysis also showed that the hydrogel microsphere vaccine dramatically amplified the intratumour density of CD103$^+$ CD11b$^-$ cDC1s, which was 2.4-, 2.3-, 2.2- and 2.0-folder higher than that of the Group1, Group2, Group3 and Group4 counterparts, respectively (Fig. 4g). The cDC1-to-cDC2 ratio was significantly elevated after the hydrogel microsphere vaccine stimulation (Fig. 4g). These results indicate that the hydrogel microsphere vaccine has a targeted amplification and activation effect on tumour-resident cDC1s. The density of tumour-resident CD103$^+$ CD11b$^-$ cDC1s but not CD24$^+$ total DCs was negatively correlated with the tumour weight in all tumour-bearing mice (Fig. 4h), suggesting the cDC1s played a critical role in combatting the tumour growth.

Tumour-resident cDC1s need to migrate into TdLNs to activate tumour-specific CD8$^+$ T cells. Flow cytometry analysis showed that the hydrogel microsphere vaccine increased the density of total MHC II$^+$ CD11c$^+$ cells in TdLNs and the spleen (Fig. 4i, j and Supplementary Fig. 11a, b), and increased the proportion of CD103$^+$ CD11b$^-$ cDC1s among total DCs from 23.3% to 60.1% in TdLNs (Fig. 4i). The density of CD103$^+$ CD11b$^-$ cDC1s in TdLNs after hydrogel microsphere vaccine

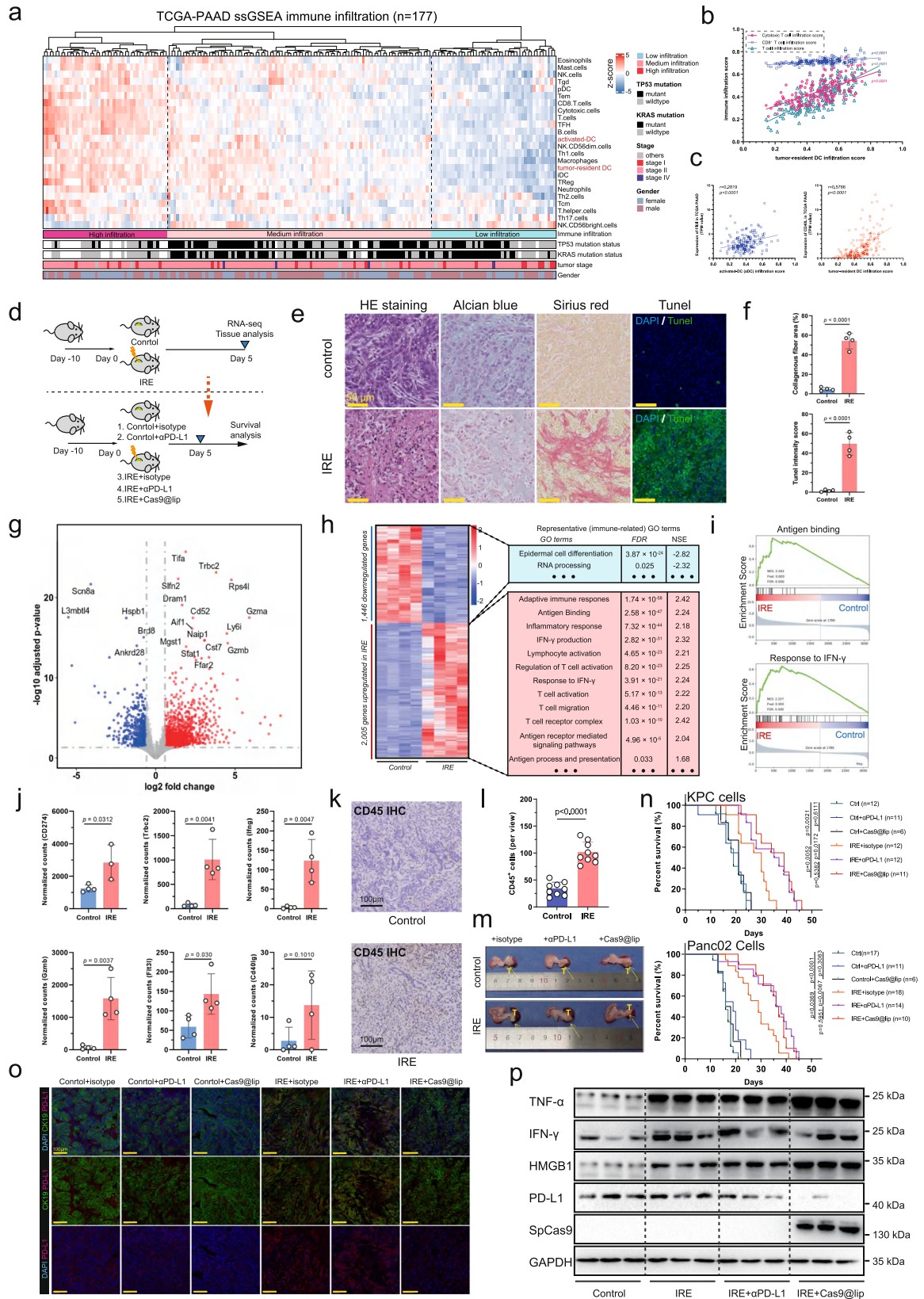

treatment was -1.5-, 1.3-, 1.3- and 1.2-fold higher than that of the Group1, Group2, Group3 and Group4 counterparts, respectively (Fig. 4k), demonstrating that the vaccine amplified the migration of tumour-resident CD103+ CD11b− cDC1s into TdLNs. Consistently, as the hydrogel microsphere vaccine stimulated the migration of CD103+ CD11b− cDC1s with highly expressed CD40 and CD86 costimulators

into the spleen (Fig. 4j, l). These results were also validated by the IF staining experiments (Fig. 4m). Moreover, the density of OVA$_{257-246}$-H-2Kb-specific tumour-resident CD103+ CD11b− cDC1s in the hydrogel microsphere vaccine-treated tumour was over fivefold higher than that in Group1-treated control mice (Fig. 4n and Supplementary Fig. 12). Taken together, these results suggest that the hydrogel microsphere

**Fig. 3 | Irreversible electroporation ablation induces immune remodelling in pancreatic cancer. a** Identification of the immune landscape of human pancreatic cancer using the bulk RNA-sequencing data of the TCGA-PAAD cohort (*n* = 177). **b** Analysis of the correlation between tumour-resident DC infiltration and T cell, CD8⁺ T cell and cytotoxic T-cell infiltration in human pancreatic cancers. **c** Analysis of the correlation between intratumour expression of Flt3lg and Cd40lg and DC infiltration in human pancreatic cancer. **d** Schematic diagram of the in vivo experiments. **e** Representative images of H&E staining, Alcian blue staining, Sirius red staining, and TUNEL staining of pancreatic cancer tissues in IRE-treated or non-IRE-treated mice. *n* = 4 biologically independent sample for each group. **f** Analysis of the necrotic and fibrotic areas of pancreatic cancer in IRE-treated or non-IRE-treated mice. *n* = 4 biologically independent samples. **g** Transcriptome profile of pancreatic cancer in IRE-treated or non-IRE-treated mice. *n* = 4 biologically independent sample for each group. The volcano plot shows the differentially expressed genes, the red dots indicate significantly upregulated genes after IRE (FDR < 0.05, $log_2$ fold change >1), and the blue dots indicate significantly down-regulated genes. **h, i** Heatmap and pre-ranked gene set enrichment analysis (GSEA) showing that the molecular pathways were significantly altered after IRE ablation therapy. **j** Determination of the expression of *CD274*, *Trbc2*, *Ifng*, *Gzmb*, *Fl3tlg* and *Cd40lg* after IRE treatment. *n* = 3 biologically independent samples. **k, l** Representative images (**k**) and statistical analysis (**l**) of IHC staining showing the infiltration of CD45-positive cells in pancreatic cancer specimens receiving IRE ablation or control tissue. *n* = 9 biologically independent samples for each group. **m** Representative images of gross tumours in mice receiving different combination therapies. **n** Survival analysis of mice bearing orthoptic tumours after different combination therapies, including control, control+αPD-L1 (200 μg, i.p. every third day), control+cas9@lip (intratumour injection), IRE+isotype (200 μg, i.p., every third day), IRE + αPD-L1 (200 μg, i.p., every third day) and IRE+cas9@lip (intratumour injection). **o** Representative images of IF staining showing the expression of CK19 and PD-L1 in the abovementioned six groups of mice bearing KPC tumours. **p** Western blotting showed the expression of Cas9 protein, HMGB1, TNF-α, IFN-γ and PD-L1 in tumours receiving different treatments. **o, p** Three independent biological samples for each group were analysed to confirm data stability. **b, c** Data are tested using Pearson's test with a two-tailed *P* value. **f, j, l** Data are presented as mean values +/− SEM. Two-tailed unpaired *t* test. **n** Survival data are analysed using Kaplan−Meier Method. Two-tailed log-rank test. Source data are provided as a Source data file.

## The hydrogel microsphere vaccine amplified CD8⁺ T-cell-specific antitumour immunity

CD8⁺ T cells have been identified as one of the most important tumour-killing immune cells. cDC1s provide MHC-I/antigen for the formation of the TCR-MHC-I-antigen complex with TCRs on CD8⁺ T cells, which is necessary for CD8⁺ T-cell-specific antitumour immunity. cDC1s also provide essential costimulators to further activate cytotoxic CD8⁺ T cells. Here, we demonstrated that the hydrogel microsphere vaccine significantly amplified the activation and proliferation of CD8⁺ T cells in TdLNs (Fig. 5a and Supplementary Fig. 13). Flow cytometry analysis showed that the density of CD8⁺ T cells in TdLNs was increased in the hydrogel microsphere vaccine-treated mice by approximately 20% compared with that in the Group1-treated control mice (Fig. 5b), and similar results were also observed in CD8- and CD4-stained TdLN specimens (Supplementary Fig. 14a, b). Notably, the density of Ki67⁺CD8⁺ T cells and IFN-γ⁺CD8⁺ T cells in the TdLNs of the hydrogel microsphere vaccine-treated mice was 2.1- and ninefold higher (*P* < 0.0001) than that in the Group1-treated mice (Fig. 5b). Further analysis showed that the density of CD103⁺ CD11b⁻ cDC1 was highly correlated with the density of total CD8⁺ T cells (Supplementary Fig. 15a), Ki67⁺CD8⁺ T cells (Supplementary Fig. 15b), and IFN-γ⁺CD8⁺ T cells (Supplementary Fig. 15c) in TdLNs, implying that the amplified CD103⁺ CD11b⁻ cDC1 cascade may have an augmented effect on the activation of CD8⁺ T cells.

We next evaluated tumour-infiltrating cytotoxic T lymphocytes (CTLs) in vivo in mice bearing orthotropic pancreatic cancer. IF staining indicated a significant increase in the density of intratumour CD8⁺ T cells in mice treated with the hydrogel microsphere vaccine (Fig. 5c). Flow cytometry analysis showed the effect of the hydrogel microsphere vaccine stimulation on the TME of pancreatic cancer, including the infiltration of T cells, NK cells and CD11c⁺Ly6G⁺ myeloid cells (Fig. 5d, e and Supplementary Fig. 16a). The flow cytometry analysis showed a minimal change in Ly6G⁺CD11b⁺ myeloid cells and NK cells across different groups (Supplementary Fig. 16b). Further analyses revealed no differences in the proportion of total F4/80⁺ macrophages in orthotopic pancreatic tumours across different treatment groups, while the M2-like immunosuppressive macrophages (CD163⁺CD80⁺ macrophages and CD206⁺CD80⁺ macrophages) slightly reduced after the CD40L and FLT3L combination treatment (Supplementary Fig. 17a, b). The density of intratumour total CD8⁺ T cells, Ki67⁺CD8⁺ T cells, IFN-γ⁺CD8⁺ T cells and Gzmb⁺CD8⁺ T cells was 2.8-,

34-, 7- and 39-fold higher than that in the Group1-treated mice, respectively (Fig. 5f and Supplementary Fig. 18). In comparison, there were no significant changes in the density of tumour-infiltrating FoxP3⁺CD25⁺ Tregs (Fig. 5f). These alterations led to a significant increase in the CD8⁺ T-cell-to-Treg ratio in pancreatic cancer (Fig. 5f), suggesting an amplification of CD8⁺ T-cell-specific antitumour immunity. The minimal change in the levels of suppressive FoxP3⁺CD25⁺ Tregs also demonstrated that the hydrogel microsphere vaccine is well targeted for cascade amplification of CD8⁺ T cells.

The intratumour and serum levels of IL12p70, IFN-γ and TNF-α (Fig. 5g−i) in tumour-bearing mice treated with the hydrogel microsphere vaccine were significantly higher than those in mice treated with counterpart's hydrogel microspheres. The levels of the liver enzymes aspartate transaminase (AST) and alanine transaminase (ALT) were not elevated in tumour-bearing mice receiving locally injected hydrogel vaccines, while AST and ALT levels were significantly increased in mice receiving intraperitoneal injection of FLT3L or αCD40 (Fig. 5j). This result suggests that the hydrogel microsphere vaccines can effectively reduce potential liver injury caused by systemic administration of cytokines. Local administration of the hydrogel microsphere vaccine did not induce an obvious immune response in the liver, kidney or lung, indicating good safety of the strategy (Supplementary Fig. 19). Taken together, these findings indicated that the hydrogel microsphere vaccine efficiently amplified CD8⁺ T-cell-specific antitumour immunity via amplification of the cDC1-mediated antigen cross-presentation cascade.

Finally, we assessed the therapeutic efficacy of the hydrogel microsphere vaccine in combination with IRE ablation for pancreatic cancer (Fig. 6a). The bioluminescence experiment demonstrated that local tumour recurrence and progression were delayed in tumour-bearing mice treated with the hydrogel microsphere vaccine (Fig. 6b, c). The median overall survival (OS) of mice bearing KPC tumours treated with the combination of hydrogel microsphere vaccine and IRE was ~30% longer than that of mice treated with IRE monotherapy, and nearly 2.5-fold longer than that of the untreated control mice (median OS in the control vs. IRE vs. IRE with hydrogel microsphere vaccine group: 19.0 vs. 36.0 vs. 48.0 days, *P* < 0.001) (Fig. 6d). Consistently, mice bearing Panc02 orthotopic tumours also significantly benefited from the hydrogel microsphere vaccine after IRE ablation, with a twofold improvement in median OS compared to that of untreated control mice (Fig. 6d). In contrast, although intraperitoneal administration of FLT3L and CD40 agonist extended survival in mice bearing orthotopic pancreatic cancer compared with ablation monotherapy, the survival benefit from local administration of the hydrogel microsphere vaccine was significantly superior (Supplementary Fig. 20a−d).

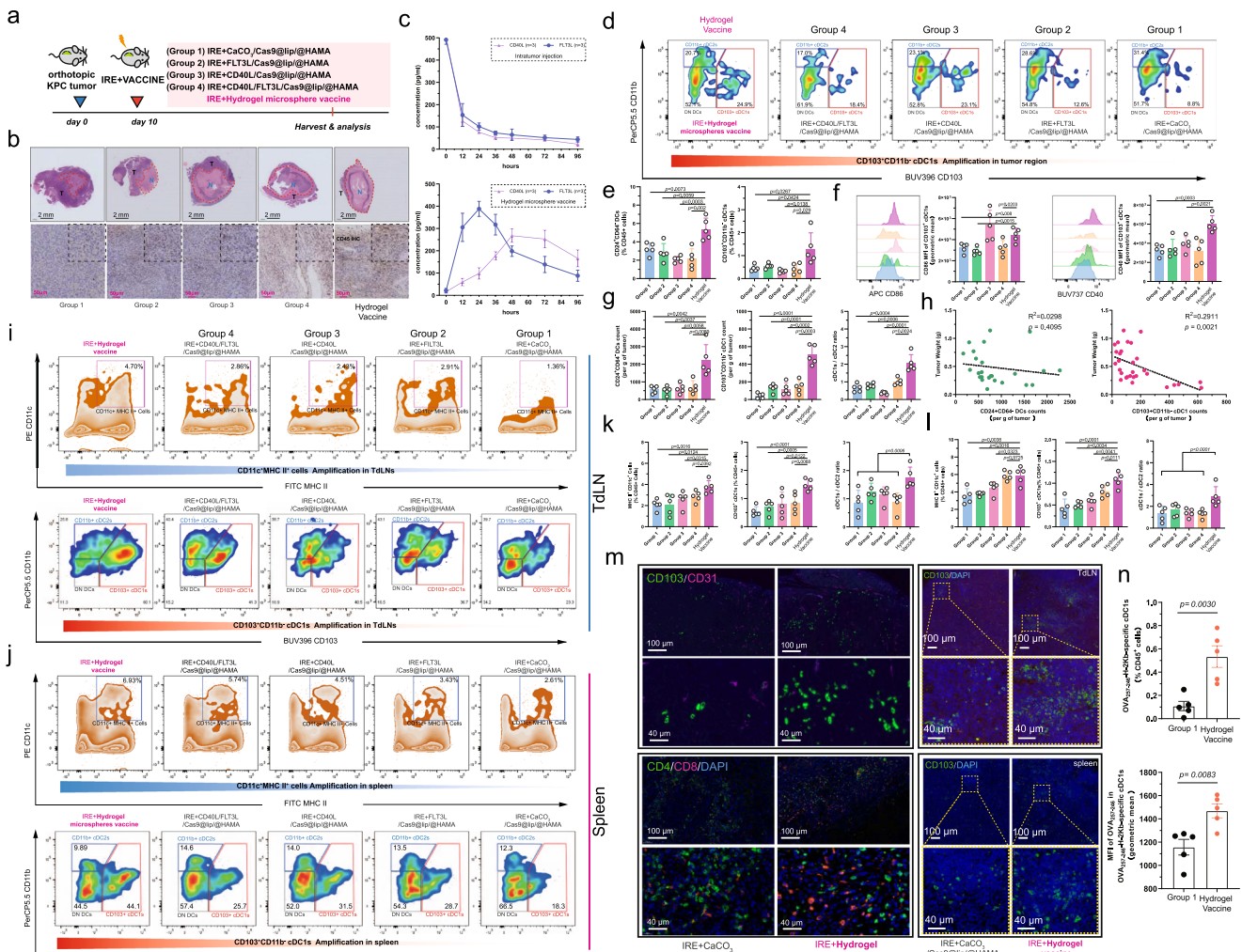

**Fig. 4 | The hydrogel microsphere vaccine amplified the recruitment, activation and migration of CD103⁺CD11b⁻ cDC1s. a** Schematic diagram of the in vivo experiments. **b** Representative images of H&E staining and CD45-labelling IHC staining of pancreatic cancer tissues after different combination therapy. Five biologically independent samples were performed to confirm data stability. **c** Analysis of the intratumour release of FLT3L and CD40L by the hydrogel microsphere vaccine or direct intratumour injection of dual cytokines. $n = 3$ biologically independent samples. **d** Representative images of flow cytometry analysis showing subpopulations of tumour-resident DCs across different groups. **e** Quantification of the proportion of tumour-resident CD24⁺CD64⁻ total DCs and CD103⁺CD11b⁻ cDC1s across different treatment groups ($n = 5$ for each group). **f** Quantification of the expression of CD86 and CD40 on the surface tumour-resident cDC1s across different groups ($n = 5$ for each group). **g** Quantification of the cellular density of tumour-resident CD24⁺CD64⁻ DCs and CD103⁺CD11b⁻ cDC1s across different treatment groups ($n = 5$ for each group). The hydrogel microsphere vaccine significantly amplified the cellular density of tumour-resident cDC1s and increased the cDC1-to-cDC2 ratio in pancreatic cancer. **h** Identification of the correlation between

the cellular density of tumour-resident total DCs or cDC1s and the tumour weight across all tumour-bearing mice. **i, j** Representative images of flow cytometry analysis showing total CD11c⁺MHC II⁺ myeloid cells and subpopulations of DCs in the TdLNs and spleens across different groups. **k, l** Quantification of the density of CD11c⁺MHC II⁺ myeloid cells and CD103⁺CD11b⁻ cDC1s in TdLNs (**k**) and spleens (**l**) in mice bearing pancreatic cancer. The hydrogel microsphere vaccine significantly amplified the density of cDC1s and increased the cDC1-to-cDC2 ratio in the TdLNs and spleens in mice bearing pancreatic cancer. **m** Representative images showing IF-stained CD103, CD31, CD4 or CD8-positive cells in pancreatic tumour tissue (left panel), TdLNs and spleens in tumour-bearing mice treated with the hydrogel microsphere vaccine or Group1 control microspheres. **n** Quantification of the density of tumour-resident OVA₂₅₇-₂₄₆-H-2Kb-specific cDC1s in hydrogel microsphere vaccine-treated mice and Group1-treated mice. **e**–**g**, **k**, **l**, **n** $n = 5$ biological independent samples for each group. Data are presented as mean values +/− SEM. Two-tailed unpaired t test. **h** $n = 25$ biological independent samples, data are tested using Pearson's test with two-tailed $P$ values. Source data are provided as a Source data file.

Next, we evaluated the systemic antitumour immune response and abscopal impact triggered by the hydrogel microsphere vaccine. The density of MHC II⁺CD11c⁺ cells and CD8⁺ T cells in the circulation of tumour-bearing mice treated with the combination of the hydrogel microsphere vaccine and IRE was significantly higher than that in mice treated with IRE monotherapy (Fig. 6e–g). The abundance of CD24⁺MHC-II⁺CD11c⁺ cells in the circulation of tumour-bearing mice treated with the combination therapy was 2.0- and 1.4-fold higher than that in control or IRE monotherapy-treated mice, respectively (Fig. 6g and Supplementary Fig. 21a). The vaccine also increased the CD8-to-CD4 ratio in the circulation (Fig. 6g and Supplementary

Fig. 21b), implying amplification of the systemic immunity of CD8⁺ T cells. The weight of distant subcutaneous metastases in mice treated with the combination of the hydrogel microsphere vaccine and IRE was only 1/5 of that in mice that received IRE monotherapy (mean weight in IRE vs. IRE combined with vaccine: 0.4860 g vs. 0.0980 g, $P < 0.0001$) (Fig. 6h, i), and the volume was 1/7 of that in mice that received IRE monotherapy (mean volume in IRE vs. IRE combined with vaccine: 596.0 mm³ vs. 87.4 mm³, $P < 0.0001$) (Fig. 6j–l and Supplementary Fig. 22a, b), indicating that the strategy of amplifying the cDC1/CD8⁺ T antitumour axis induced a strong abscopal effect.

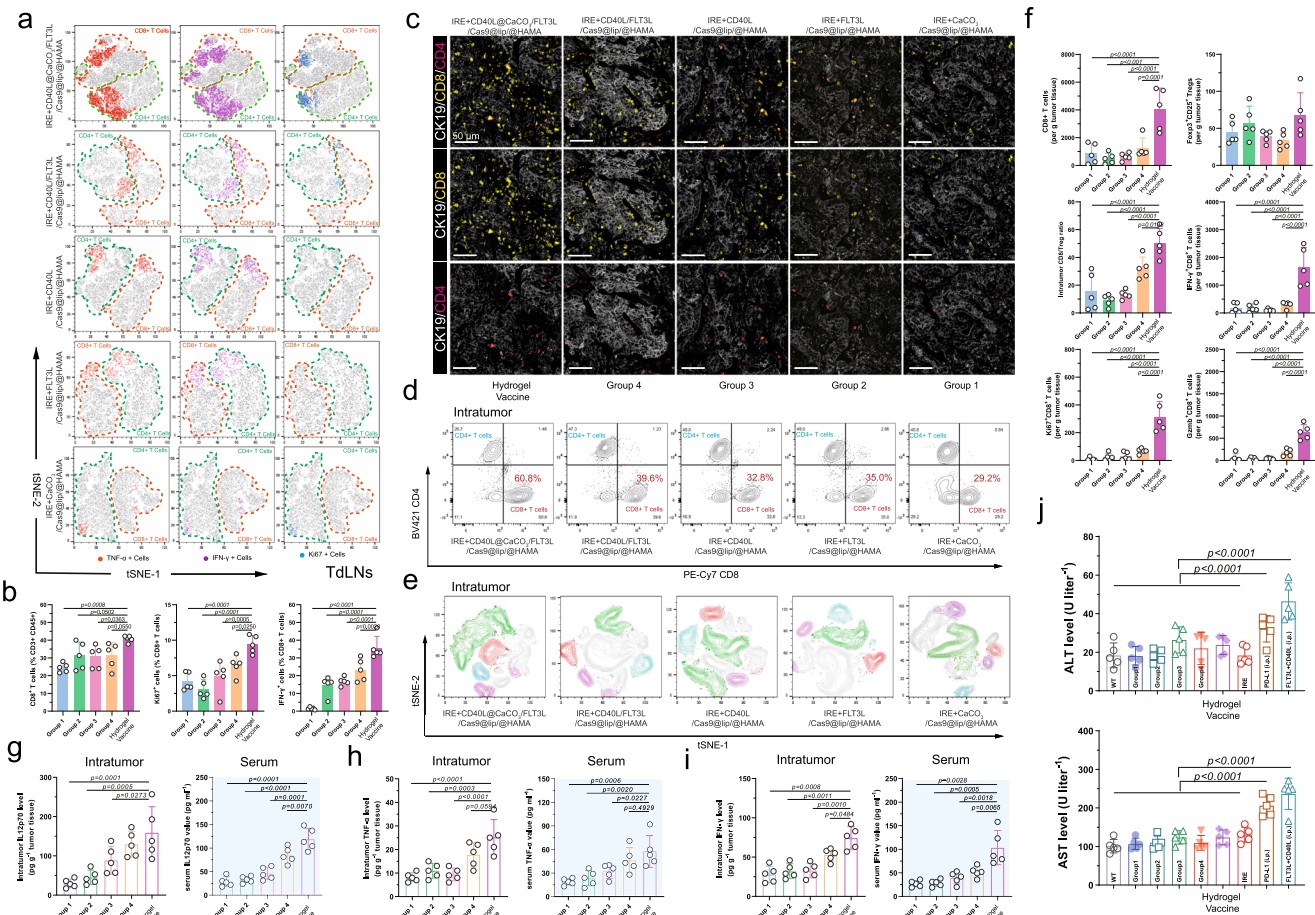

**Fig. 5 | The hydrogel microsphere vaccine amplified CD8[+] T-cell-specific anti-tumour immunity. a** Representative images showing the t-SNE analysis of CD8, CD4, IFN-γ, TNF-α and Ki67 markers in TdLNs across different treatment groups. **b** Identification of the proportion of CD8[+] T cells (left), Ki67[+]CD8[+] T cells (middle) and IFN-γ[+]CD8[+] T cells in the TdLNs. **c** Representative images showing the IF multicolour-labelled CD8-, CD4- and CK19-positive cells in pancreatic tumour tissue in tumour-bearing mice treated with the hydrogel microsphere vaccine or control hydrogel microspheres. Five biologically independent samples were performed to confirm data stability. **d** Representative images of flow cytometry analysis showing the proportion of tumour-infiltrating CD8[+] T cells and CD4[+] T cells across different treatment groups. **e** Representative images showing the t-SNE analysis of the distribution of tumour-infiltrating NK cells (NK-1.1[+]CD45[+], purple), total myeloid cells

(CD11c[+]Ly6G[+] CD45[+], green), CD8[+] T cells (CD8[+]CD3[+]CD45[+], red) and CD4[+] T cells (CD4[+]CD3[+]CD45[+], blue) across different treatment groups. (**f**) Identification of the cellular density of tumour-infiltrating CD8[+] T cells (top left), FoxP3[+]CD25[+] Tregs (top right), IFN-γ[+]CD8[+] T cells (middle right), Ki67[+]CD8[+] T cells (bottom left) and Gzmb[+]CD8[+] T cells (bottom right) across different treatment groups. **g–i** Quantification of intratumour and serum levels of IL12p70 (**g**), IFN-γ (**h**), and TNF-α (**i**) level. **j** Identification of serum ALT and AST values. **b**, **f**, **g–i** *n* = 5 biological independent samples for each group. Data are presented as mean values +/− SEM. Two-tailed Dunnett *t* test. **j** *n* = 5 biological independent samples for each group. Data are presented as mean values +/− SEM. Two-tailed unpaired *t* test. Source data are provided as a Source data file.

We also investigated which immune cells were responsible for the abscopal effect. Based on the function of hydrogel microsphere vaccine, we focused on DCs, macrophages and CD8[+] T cells. The results showed no difference in the proportion of these immune cells in the subcutaneous distant tumours across different groups (Fig. 7a). Subsequently, we queried whether the function of CD8[+] T cells was altered. The results showed a significant increase in the proportion of IFN-γ[+]CD8[+] T cells in subcutaneous tumours in mice receiving IRE combined with vaccine therapy (Fig. 7b, c). This result indicates that the activated CD8[+] T cell may be responsible for the abscopal effect. This is consistent with the result that the rational design of the hydrogel vaccine to amplify the cDC1/CD8[+] T-cell axis. The abscopal effect was further confirmed in models with lung metastases as well as liver metastases (Fig. 7d). Notable, the abscopal effect showed a strong specificity for pancreatic cancer lineage, while the lung (KL) and liver cancer (Hepa1-6) cells induced distant metastasis did not response to the local combination therapy (Fig. 7e, f and Supplementary Fig. 22c, d). Further, we performed rescue experiment to evaluate the effect of depletion of CD8 or blockade of MHC-I on the antitumour abscopal

effect (Fig. 8a). Single dose intraperitoneal administration of antibody drugs could achieve CD8[+] T-cell depletion in circulation, and MHC-I blockade on cDC1s infiltrating in TdLNs (Fig. 8b). The results showed that after treatment with IRE and hydrogel vaccine for subcutaneous tumour, CD8[+] T-cell depletion or MHC-I blockade significantly promoted the growth of distant subcutaneous metastases (Fig. 8c). The CD8[+] T-cell depletion or MHC-I blockade also promoted lung (Fig. 8d, e) and liver (Fig. 8f, g) metastases of Panc02 cells after combination therapy for the orthotopic Panc02 tumours. These results enhanced the conclusion that the abscopal effect is a highly targeted CD8[+] T-cell-mediated adaptive antitumour effect.

## Discussion

Here, we develop a cancer hydrogel microsphere vaccine for post-ablation immunotherapy. In contrast to recent work focusing on intratumour delivery of exogenous antigens or immunoadjuvants, the hydrogel microsphere vaccine acts as a general immune amplifier to trigger a rocket-like amplification of the CD103[+] CD11b[−] cDC1-mediated antigen cross-presentation cascade, resulting in dramatic

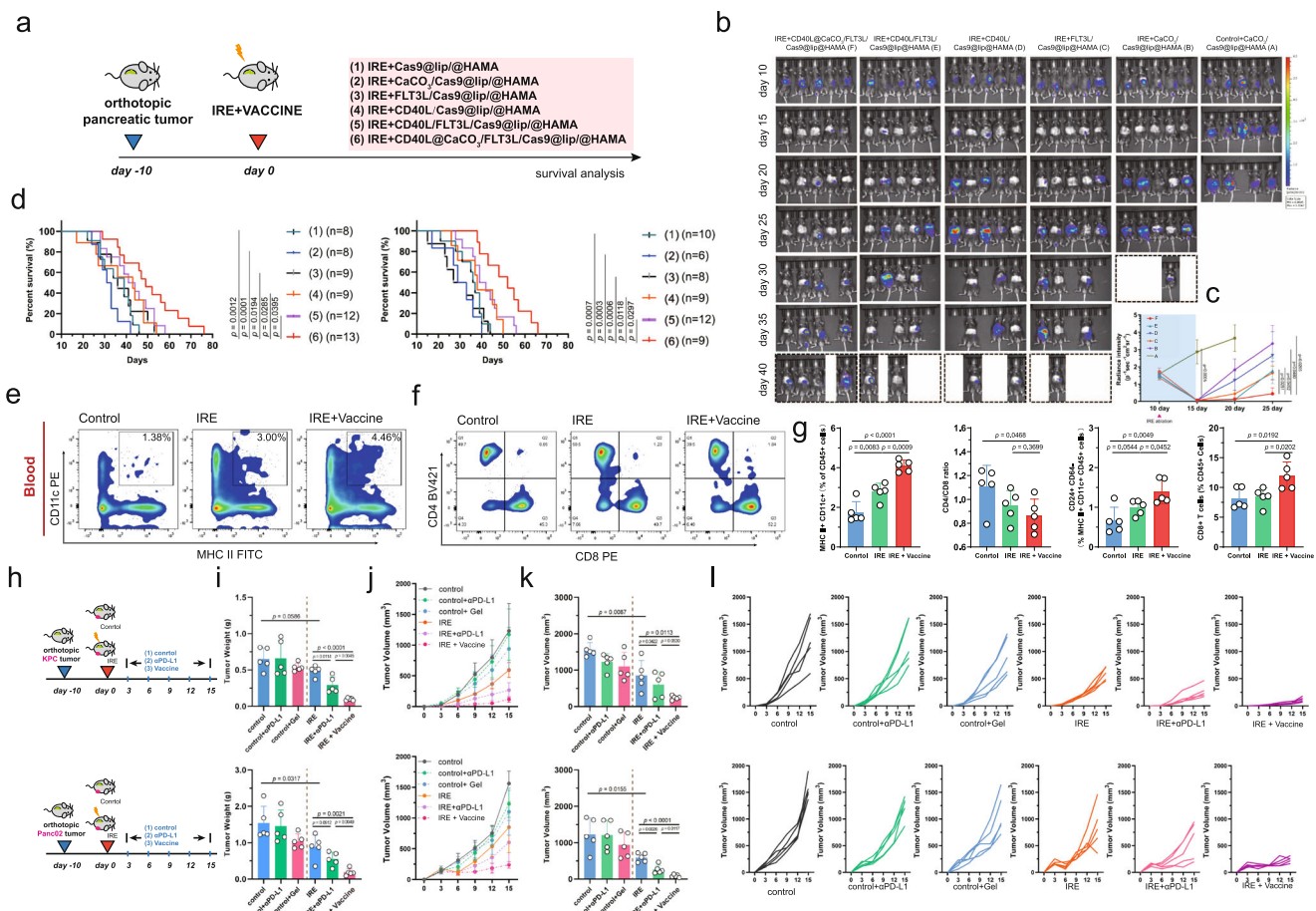

**Fig. 6 | Therapeutic efficacy of the hydrogel microsphere vaccine in combination with IRE ablation for pancreatic cancer. a** Schematic diagram illustrates different therapeutic combinations for in situ immune modulation.
**b, c** Representative images of the bioluminescence experiment demonstrating local tumour recurrence and progression in tumour-bearing mice treated with the hydrogel microsphere vaccine or different control hydrogel microspheres (**b**). Analysis of the radiance intensity indicating significantly attenuated tumour recurrence in hydrogel microsphere vaccine-treated mice (**c**). The sample size in (**c**) corresponds to that in (**b**). Data are presented as mean values +/− SEM. Two-tailed unpaired $t$ test. **d** Survival analysis of mice bearing orthoptic tumours after different combination therapies. The therapeutic strategies and sample size of each group are labelled in the figure. Survival data are analysed using Kaplan–Meier Method. Two-tailed log-rank test. **e, f** Representative images of flow cytometry analysis showing the proportion of circulating CD11c⁺ MHC II⁺ cells (**e**) and CD8⁺ T (**f**) cells in

control, IRE monotherapy-treated and IRE+ hydrogel microsphere vaccine-treated mice. **g** The hydrogel microsphere vaccine significantly amplified the cellular density of circulating CD11c⁺MHC II⁺ cells (left), CD8⁺ T-cell-to-CD4⁺ T-cell ratio (middle left), cellular density of CD24⁺CD64⁻ cells (middle right) and cellular density of CD8⁺ T cells (right) in KPC tumour-bearing mice. **h** Schematic diagram showing the in vivo experiments for determination of the abscopal effect. The upper panel indicates experiments performed using mice bearing KPC orthotopic pancreatic tumours. The lower panel indicates experiments performed using mice bearing Panc02 orthotopic pancreatic tumours. **i, j** Comparison of the weight (**i**) and volume (**j, k**) of distant metastatic tumours across different groups. **l** Line chart showing the growth of individual tumours in each treatment group. **g, i, j, k** $n = 5$ biologically independent samples for each group. Data are presented as mean values +/− SEM. Two-tailed unpaired $t$ test. Source data are provided as a Source data file.

---

amplification of the antitumour immunity of endogenous CD8⁺ T cells after ablation therapy. Notably, our data show that the immune amplification induced by the hydrogel microsphere vaccine dramatically inhibited local recurrence and distant metastasis formation after tumour ablation therapy, promising to provide benefits for pancreatic cancer patients.

Targeting antigen cross-presentation holds great potential for antitumour immunotherapy[49]. However, current strategies, including artificial antigen cross-presentation systems[50] or immunoadjuvant delivery systems, are constrained by inadequate antitumour immune responses, making it challenging to completely eradicate established solid tumours[51–53]. Here, we developed a hydrogel microsphere vaccine as a general immune amplifier for cancer immunotherapy. The hydrogel microsphere vaccine enhances in situ recruitment and amplification of tumour-resident CD103⁺ CD11b⁻ cDC1s by rapidly releasing FLT3L, followed by the controlled release of CD40 agonist in the acidic TME, which further amplified activation and antigen-priming

of cDC1s. The immune amplification induced by the hydrogel microsphere vaccine ultimately promoted a robust CD8⁺ T-cell response while having negligible immunological side effects. Recently, Jiang et al.[37] reported that MEK and autophagy co-inhibition coupled with αCD40 invoked immune re-polarisation and was an attractive therapeutic approach for pancreatic cancer immunotherapy. However, the moderate loss of cDC1s induced by the systemic administration of αCD40 may impair antigen presentation to CD8⁺ T cells and attenuate the expansion and activation of CD8⁺ T cells. The application of hydrogel microsphere system to promote cascade amplification of cDC1-driven CD8⁺ T-cell antitumour immunity amplification addressed the problem of insufficient immune reactivation. And this highly targeted immune amplification via cDC1/CD8⁺ T-cell axis also addresses two long-standing issues in the field of antitumour nanovaccines, including the safety concerns associated with potential off-target and pan-immune remodulation, as well as the high cost and side effects associated with systemic drug delivery[54].

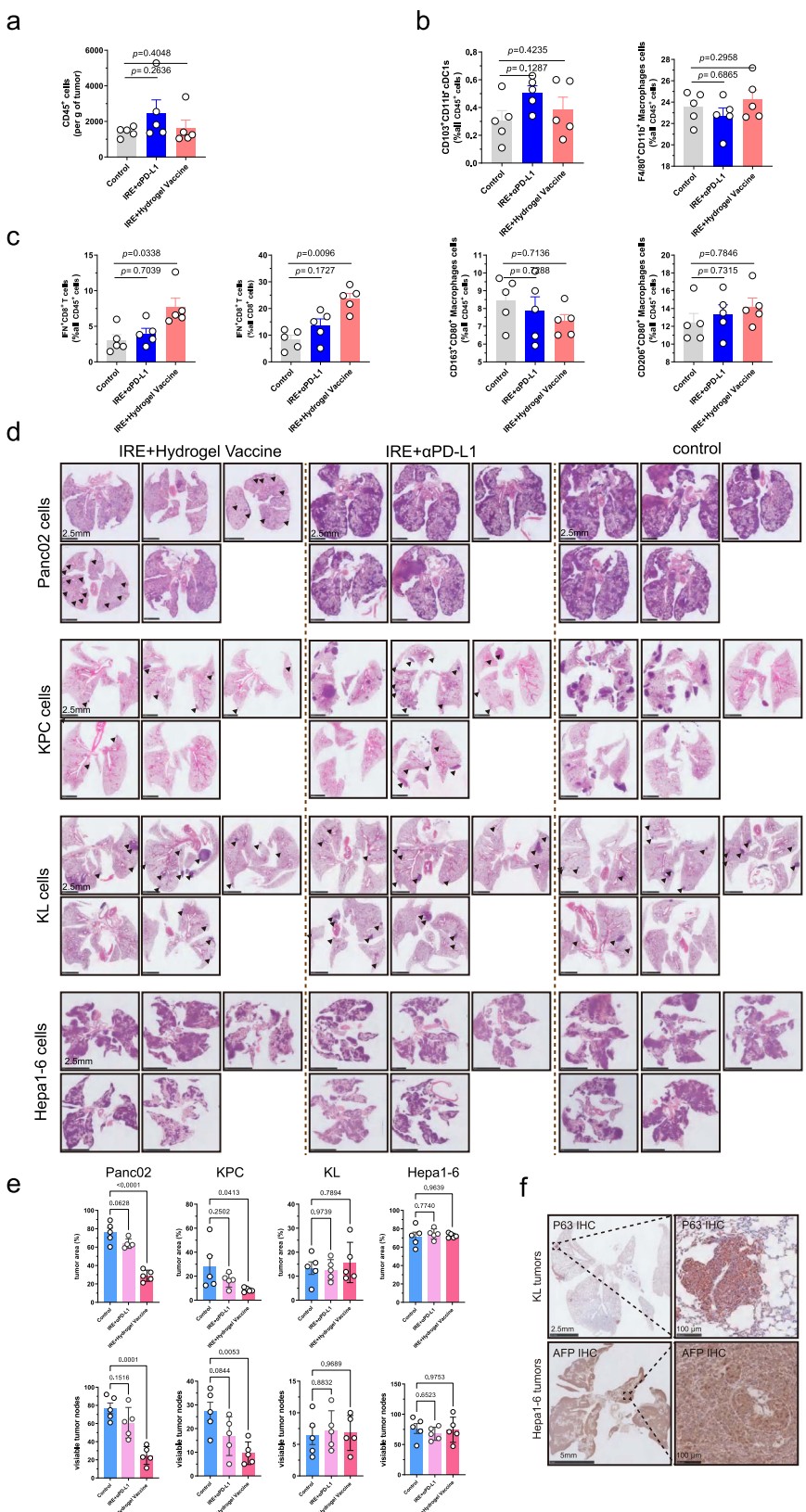

Notably, the hydrogel microsphere vaccine synergises effectively with the biological response elicited by ablation therapy. The hydrogel microsphere vaccine activates CD103$^+$ CD11b$^-$ cDC1s for endogenous antigen cross-presentation by taking up necrotic debris released upon tumour ablation therapy[55]. Endogenous TAA-educated CD8$^+$ T cells can more effectively and specifically target cancer cells and overcome tumour heterogeneity. In recent years, low-dose radiation or ICD chemotherapy has also attracted significant attention for improving tumour immunotherapy by promoting proinflammatory factor and antigen release[3,56,57]. The hydrogel microsphere vaccine can also synergise well with these ICD-based therapeutics, making it a more general local immunotherapy strategy for solid tumours. Finally, we

**Fig. 7 | Activated CD8⁺ T cells play a key role in the combination therapy-induced abscopal effect. a** Flow cytometry analysis revealed no differences in the count of tumour-infiltrating CD45⁺ cells. **b** Flow cytometry analysis revealed no differences in proportion of cDC1s, M1 and M2 cells in distant subcutaneous tumours across different treatment groups. **c** Flow cytometry analysis showed that the proportion of IFN-γ⁺CD8⁺ T cells in distant tumours increased significantly in Panc02 subcutaneous tumours. **d** Representative images of H&E staining of lung metastasis of Panc02, KPC, KL and Hepa1-6 cells across different treatment groups.

**e** The hydrogel microsphere vaccine combined with IRE ablation significantly reduces the number of tumour nodes and tumour area in KPC and Panc02-induced lung metastases. **f** Representative images of IHC staining showed the expression of P63 in KL cells and the expression of AFP in Hepa1-6 cells. Three biologically independent samples were performed to confirm data stability. **a–c**, **e** $n = 5$ biologically independent samples. **a–c, e** $n = 5$ biologically independent samples. Data are presented as mean values +/− SEM. Two-tailed Dunnett $t$ test. Source data are provided as a Source data file.

demonstrated that the hydrogel microsphere vaccine combined with ablation therapy dramatically increased survival in a series of highly clinically relevant tumour models. Given the current dilemma in immunotherapy of "cold" solid tumours, the local immune amplification strategy holds great promise for translation into clinical application.

The limitations of the study should also be emphasised. The complex drug formulation increases the difficulty of mass production and clinical translation. Simplification of drug delivery vehicles and the support of subsequent pharmaceutical technologies are necessary for subsequent clinical transformation. In the rescue issues, we used MHC-I blockade instead of targeted cDC1 depletion due to current technical limitations. This method could not completely exclude the possible activation of CD8⁺ T cells by other APCs. Subsequent studies should also focus on the upstream activation pathways of CD8⁺ T cells in immune activation.

In conclusion, the hydrogel microsphere vaccine provides a strategy to efficiently amplify the antitumour immunity of endogenous CD8⁺ T cells through the promotion of the cDC1-mediated immune cascade. Compared to conventional ICB therapies, the combination strategy may also be an emerging trend in the intratumour immunotherapy in the future.

## Methods

### Ethics statement

The animal experiments were approved by the Committee of Animal Rights and Welfare at Ruijin Hospital Shanghai Jiao Tong University School of Medicine and Committee of Animal Rights and Protection at CAS Centre for Excellence in Brain Science and Intelligence Technology, Shanghai. The mice were kept in the SPF barrier. The facility environment was as follows: Temperature was 20–26 °C, and daily temperature difference was less than 4 °C. Humidity was 40–60%. The frequency of air change is 15–20 times per hour, and the light and dark alternates every 12 h. The maximum noise is less than 60 dB. According to the Ethics Committee, the size of subcutaneous tumours should not exceed 2000 mm³. The experiment should be terminated in time when the tumours develop ulceration, necrosis, or server infection. The mice show obvious cachexia should be euthanized. In survival analysis, in addition to the death outcome, the moribund status of mice, including lack of response to stimulation, immobility, or inability to eat or drink, were also used as an endpoint of euthanasia. For euthanasia, $CO_2$ (1.5–2 L/min) was pumped at a constant rate into sealed cages containing mice to sacrifice the mice. Finally, the mice were confirmed dead or treated with cervical dislocation.

### Preparation of the hydrogel microsphere vaccine

**CRISPR guide-RNA design and plasmid construction.** The guide-RNAs targeting exon regions of mouse PD-L1 gene (CD274) were designed using the Benching Webtool (https://www.benchling.com/; sgRNA-1: 5′-AGTATGGCAGCAACGTCACGA-3′; sgRNA-2: 5′-CGGCTC CAAAGGACTTGTACG-3′; sgRNA-3: 5′-CTGCTGCATAATCAGCTACG G-3′; sgRNA-4: 5′-GCATAATCAGCTACGGTGGTG-3′; sgRNA-5: 5′-AGA CGTCAAGCTGCAGGACGC-3′). The gRNAs were constructed into the pX330 plasmid (Addgene:42230) plasmid using the BbsI enzyme. The high-content endotoxin-free plasmid was extracted using the Endotoxin-Free Giga Plasmid Purification Kit (Thermo Fisher, Cat. # A31233).

**Preparation of the CD40L@CaCO₃ nanoparticles.** The CD40L@CaCO₃ nanoparticles were prepared by chemical precipitation of $Ca^{2+}$, $CO_3^{2-}$ and mouse CD40 Ligand (R&D, Cat. #8230-CL-050/CF), respectively. Briefly, 1 ml volume of 100 mM CaCl₂ Tris-HCl buffer solution (1 mM) was supplied with 1000 µg CD40L and 10 mg HOOC–PLGA-PEG–PLGA-COOH block polymer, and the mixed system was further mixed with 10 mM of NaCO₃ HEPES buffer solution (50 mM). The mixed solution was stirred at 4 °C for 12 h. Finally, the *CD40L@CaCO₃* nanoparticles were collected by centrifugation (12,000×*g*, 5 min) at 4 °C.

**Preparation of the Plasmid@Lip nanoparticles.** To prepare the Plasmid@Lip nanoparticles, methanol solution of 10 mM DLin-MC3-DMA (Absin Biotech, Cat. #abs820381)/Cholesterol/DSPC/PEG₂₀₀₀-DSPE at a molar ratio of 52/38.5/8/1.5 and 1×PBS solution containing Cas9 plasmid were mixed using a microfluidic device at a flow of 10 mL/min with a flow rate ratio of 3:1 (aqueous to lip phase). The mixed solution was left standing for 30 min at room temperature. The mixture was then transferred into a dialysis bag (interception molecular weight: 3500 Da) for 24 h of dialysis. The plasmid-to-lip ratio (w/v ratio) is 1:1, 1:1.5, 1:2, 1:2.5 and 1:3 according to the experimental design.

**Preparation of the hydrogel microsphere vaccine (CD40L@CaCO₃/FLT3L/Cas9@Lip/@HAMA) and counterparts.** The prepared CD40L@CaCO₃ and Cas9@Lip were added to the HAMA/FLT3L aqueous solution, resulting in a concentration of 2 wt% HAMA, 0.1 wt% LAP blue initiator, 500 µg/mL FLT3L and ~500 µg/mL CD40L. The plasmid-to-lip ratio (w/v) used in the final hydrogel system was 1:1, and the concentration of Cas9 plasmid was 50 ng/µl in the final system. The hydrogel solution was used as microfluidic water, and a 5 wt% SPAN80 paraffin solution was used as an oil phase to prepare the drug-loaded hydrogel microspheres vaccine by microfluidic device. Meanwhile, four kinds of counterparts were also synthesised as control vaccines. The name and composition of the control vaccines are listed as follow. Group1 represented the CaCO₃/Cas9@Lip/@HAMA counterpart; Group2 represented the FLT3L/Cas9@Lip/@HAMA counterpart; Group3 represented the CD40L/Cas9@Lip/@HAMA counterpart; Group4 represented the CD40L/FLT3L/Cas9@Lip/@HAMA counterpart. To make the figures more concise, names are used in the bar charts to represent these control vaccines. We used "hydrogel microsphere vaccine" or "hydrogel vaccine" represented for the final assembled hydrogel microsphere vaccine.

### Cell culture

The Panc02 cells were a gift from Dr. Wei Tao (Department of General Surgery, The First Affiliated Hospital, Zhejiang University School of Medicine, Hangzhou, China). The Hepa1-6 cell line (Cat. # SCSP-512) was bought from the National Collection of Authenticated Cell Cultures (Shanghai, China). The KPC cells were derived from 4 to 6-month-old male *Kras*^LSL-G12D^; *Tp53*^fl/+^; *Pdx1*-Cre (KPC) mice. The KL cells were derived from 12 to 16-month-old male mice (*Kras*^LSL-G12D^; *Stk11*^fl/fl^, AAV9-Cre inhaled).The *Kras*^LSL-G12D^ strain (C57BL/6-*Kras*^em4(LSL-G12D) Smoc^, stock number: NM-KI-190003, RRID: IMSR_NM-KI-190003), *Tp53*^fl/fl^ strain (C57BL/6-*Tp53*^tm2Smoc^, stock number: NM-CKO-18005, RRID: IMSR_NM-CKO-18005), *Pdx1*-Cre strain (C57BL/6JSmoc-Pdx1^em(2A-Cre)1Smoc^, stock number: NM-KI-225082, RRID: IMSR_NM-KI-225082), and *Stk11*^fl/fl^ strain

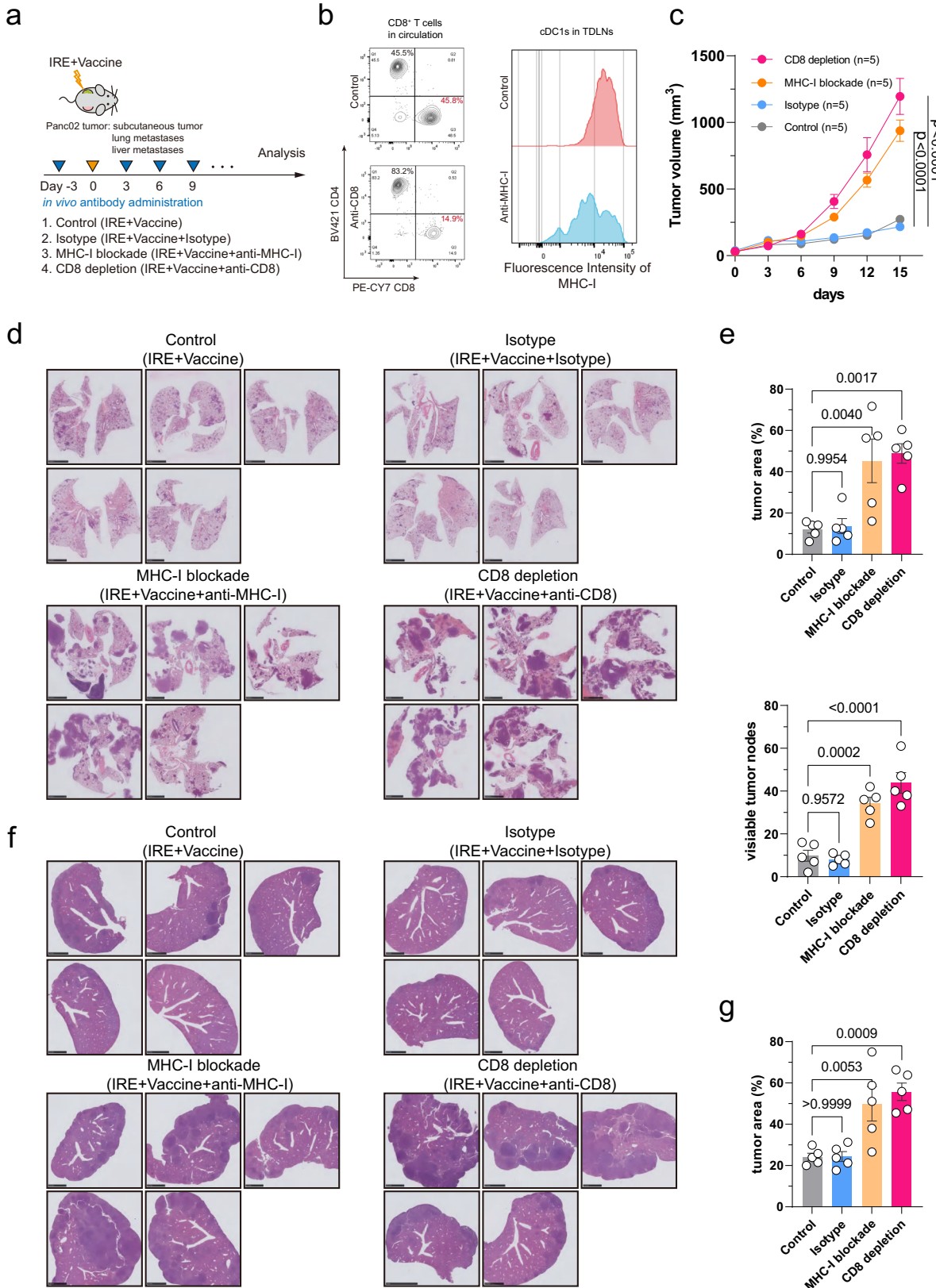

(C57BL/6J^{Smoc-Stk11tm(flox)1Smoc}, stock number: NM-CKO-18014, RRID:IMSR_NM-CKO-18014) were obtained from Shanghai Model Organisms. Briefly, the pancreatic tissue from the KPC mouse was dissociated using the Mouse Tumour Dissociation kit (Miltenyi Biotech, Cat. #30-096-730) according to the manufacturer's protocols. The CD45^−EpCAM^+ (BioLegend, Clone: G8.8, Cat. #118208;

eBioscience, Cat. #47-0451-82) alive cells were isolated by FACS sorting for long-term culture. Ovalbumin (OVA)- or Renilla luciferase (RLUC)-overexpressing lentiviruses containing the puromycin selection marker was used to generate OVA- (KPC-OVA, Panc02-OVA) or LUC- (KPC-LUC, Panc02-LUC) stably-expressed pancreatic cancer cells, respectively. The cancer cells were cultured in DMEM medium (Sigma,

**Fig. 8 | In vivo depletion of CD8$^+$ T cells or blockade of MHC-I impairs the abscopal effect induced by IRE and hydrogel vaccine combination therapy.** **a** Schematic diagram of the in vivo experiments. **b** Flow cytometry analysis demonstrated the efficiency of CD8 depletion and MHC-I blockade in mice bearing pancreatic tumours. **c** Comparison the volume of distant metastatic tumours across different groups. **d** Representative images of H&E staining of lung metastases of Panc02 cells across different treatment groups. **e** In vivo depletion of CD8$^+$ T cells or blockade of MHC-I significantly increases lung metastasis of Panc02 cells after the combination therapy of IRE and hydrogel vaccine. **f** Representative images of H&E staining of liver metastasis of Panc02 cells across different treatment groups. **g** In vivo depletion of CD8$^+$ T cells or blockade of MHC-I significantly increases liver metastasis of Panc02 cells after the combination therapy of IRE and hydrogel vaccine. **c**, **e**, **g** $n$ = 5 biologically independent samples. Data are presented as mean values +/− SEM. Two-tailed Dunnett $t$ test. Source data are provided as a Source data file.

Cat. #RNBK3090) containing 10% FBS and 10 units/L penicillin−streptomycin at a standard cell incubator (5% $CO_2$). The molecular feature markers of the Panc02 and KPC markers are presented in Supplementary Fig. 23a. In vitro analysis showed that the Cas9 plasmid system efficiently inhibited the expression of PD-L1 on pancreatic cancer cells (Supplementary Fig. 23b).

## In vitro identification of differentiation and maturation of BMDCs

Induction and identification of the differentiation and maturation of BMDCs were performed according to a standard protocol reported by Inaba et al. with minor modifications[58]. The bone marrow was collected from the tibia and femur of 6-week male C57BL/6J mice. Then, the bone marrow cells were incubated with 1× RBC lysis buffer (Invitrogen, Cat. #00-4333-57) for the removal of red cells. Next, the bone marrow cells were collected by centrifugation (500×$g$ for 5 min) and were washed three times using ice-cold 1×PBS. The purified BMDCs were seeded into a 24-well plate with 10% FBS RIPM-1640 medium ($1 \times 10^6$ cells per well), and were cultured in a standard incubator with a humidified atmosphere and 5% $CO_2$. For in vitro induction of BMDCs, the culture medium was supplied with murine GM-CSF (20 ng/ml, PeproTech, Cat. #315-03-20) and murine IL-4 (10 ng/ml, PeproTech, Cat. #214-14-5). On days 2 and 4, ~750 µl of the top layer feeding medium was absorbed, and an equal amount of fresh medium was carefully added into each culturing chamber. On day 6, suspended and semi-adherent DCs in the feeding medium were collected by centrifugation (500×$g$ for 5 min). The pre-mature DCs were reseeded into a 24-well plate with feeding medium containing GM-CSF and IL-4 ($1 \times 10^6$ cells per well). The mouse CD40 Ligand (R&D, Cat. #8230-CL-050/CF), cytokines FLT3L, or the hydrogel microspheres were added to induce the maturation and differentiation of the BMDCs. The cells were harvested after 48 h and processed for flow cytometry analysis.

## Animal experiments and in vivo irreversible electroporation

Male C57BL/6 mice (C57BL/6J-Slac, 6 weeks old) were purchased from Shanghai Lingchang Laboratory Animal Centre. For the establishment of orthotopic pancreatic cancers, a surgical incision was made in the left abdomen of the mouse to expose the pancreas tail, and 50 µl of diluted Matrigel (Corning, Cat. #354248) containing $1 \times 10^6$ of cancer cells (KPC, Panc02, KPC-LUC, or KPC-OVA cells according to experimental design) was injected into the tail of the pancreas. Prewarmed 1× PBS was used to wash the injection site 3 times to accelerate Matrigel coagulation and reduce $2 \times 10^6$ cancer cells were injected into the flank of the mice. The irreversible electroporation treatment was performed 10 days after the orthotopic tumour injection. The mice were anaesthetised, and the orthotopic pancreatic tumour was exposed by surgical incision. Each tumour (about 5 mm in length) was treated three times with IRE ablation. The in vivo IRE treatment was performed using the BTX ECM 830 Electro Square Porator Electroporator (BTX, Cat. # ECM 830, Harvard Bioscience), and the IRE parameters were set as follows: electric intensity 1750V/cm; frequency 1 Hz; wavelength 100 µs; pulse count 99. To minimise the impact of surgical procedures by different people, all the in vivo IRE interventional ablations were performed by X.L. and Y.C. Approximately 20–30 µl of hydrogel microsphere vaccine or control hydrogel microspheres were injected into the tumour tissue using Nanoject III (Drummond, Cat. # 3-000-207) after the IRE ablation.

## Evaluation of tumour growth and mouse survival

Mice bearing orthotopic pancreatic tumours (KPC-OVA, Panc02-OVA, KPC-LUC, or Panc02-LUC) were randomly assigned to different treatment groups, receiving Hydrogel Microsphere Vaccine, Group1, Group2, Group3 or Group4 injections (50–80 µl per injection) after IRE ablation. To evaluate the abscopal effect, $2 \times 10^6$ cancer cells were injected subcutaneously 4 days before the orthotopic IRE ablation to ensure the formation of subcutaneous tumours. Body weight and tumour volume were evaluated every 3 days. To evaluate the subcutaneous tumour volumes, the maximum and minimum axis of the subcutaneous tumours were evaluated using a calliper after the mice were isoflurane-anaesthetised. The estimated tumour volume ($V$) was calculated using the following formular: $V = 0.52 \times L \times W^2$ ($L$ = the major tumour axis; $W$ = the minor tumour axis). For all experiments, mice that died within 5 days of surgery were considered surgical failures, and were not included in the follow-up study. The experiments were performed under the guidelines of Laboratory Animals Welfare and Ethics of the Institute.

## Administration of antibody drugs in vivo

For in vivo immunotherapies, the anti-PD-L1 antibody (BioXCell, Clone: 10 F.9G2, Cat. #BP0101) or the isotype control (BioXCell, Clone: MOPC-21, Cat. #BP0090) were injected intraperitoneally every third day (200 µg per injection, i.p.). For in vivo safety assessment, the anti-CD40 antibody (100 µg per injection, i.p.; BioXCell, Clone: FGK4.5, Cat. # BP0016-2) were injected every third day for three times, and the FLT3L (30 µg per injection, i.p.; Sino Biological, Cat. #51113-M02H) were injected every day for 9 continuous day as previously reported. CD8$^+$ T depletion or MHC-I blockade was achieved by intraperitoneal injection of anti-CD8β antibody (InVivoMAb anti-mouse CD8β, Clone:58-5.8, Cat. #BE0223) or anti-MHC-I antibody (InVivoMAb anti-mouse MHC Class I H-2Kb, clone Y-3, Cat. #BE0172). According to experimental design, 200 µg of anti-CD8β, anti-MHC-I antibody or Isotype antibody was administered intraperitoneally on days −3, 0, 3 then every three days until the observation endpoint after the IRE and Hydrogel Vaccine combination therapy.

## Generation and assessment of the lung and liver metastasis models

The lung metastasis models of different cancers were established using tail-vein injection. To generate the lung metastasis model, $1 \times 10^7$ cancer cells were diluted in 200 µl saline, and the mixture was injected into the tail vein smoothly. The liver metastasis models of different cancers were established using the semi-spleen method. The spleens of the mice were first surgically exposed, and then the spleen was clamped from the middle with a 4 mm mini-vascular clip, avoiding damage to the blood vessels on both sides of the spleen. A total of 200 µl saline containing $1 \times 10^7$ cancer cells were steadily injected into the lower half of the spleen, followed by 200 µl of saline for perfusion and irrigation. The vessels at the hilum of the lower part of the spleen were ligated, and then the lower part of the spleen was cut off. The spleen was then inserted back into the abdominal cavity. Finally, the skin was sutured layer by layer. To test the potential abscopal effect induced by different therapies, the tumour-bearing were first treated with IRE, IRE combined with PD-L1 or IRE combined with the hydrogel microsphere vaccine. The different tumour cell lines were then

immediately injected into the mice via the tail-vein injection or the semi-spleen method. The mice were followed up and sacrificed after 2 weeks. The lungs or the livers were harvested for further experiments and analysis.

## The bioluminescence experiments

The in vivo bioluminescence experiments were performed using mice bearing LUC-labelled KPC orthotopic pancreatic tumours. The tumour-bearing mice were anaesthetised with isoflurane inhalation, and were injected with Luciferase substrate (150 mg/kg, i.p., D-Luciferin, BioVision, Cat. #7002-1). After 10 min of injection, the mice were put into the IVIS spectrum imager (IVIS Lumina III, PerkinElmer, CLS136334) for bioluminescence analysis. The radiance intensity of region of interest was assessed using the Living Image Software (Living Imaging, PerkinElmer, version 4.2). The bioluminescence experiments were performed at the Department of Laboratory Animals, Shanghai Jiao tong University School of Medicine. The experiments were performed under the guidelines of Laboratory Animals Welfare and Ethics of the Institute.

## Tumour digestion, dissociation and isolation of tumour-infiltrating lymphocytes (TILs)

The tumour was digested using the Mouse Tumour Dissociation Kit (Miltenyi Biotech, Cat. #30-096-730) according to the manufacturer's protocols. Briefly, the tumour tissue was washed three times using ice-cold 1×PBS. Next, the tumour specimens were cut into small pieces (2 × 2 mm), and transferred into the MACS C Tubes (Miltenyi Biotech, Cat. #130-093-237) containing the Miltenyi digestion enzyme cocktails. The MACS tubes were loaded into the gentleMACS Octo Dissociation with Heaters (Miltenyi Biotech, Cat. #130-096-427) for tumour digestion and dissociation (programme: 37 °C_m_TDK_2). After the digestion, 2.5 ml of ice-cold DMEM medium containing 10% FBS was added into each MACS tube to end the enzymatic digestion. The undigested tissue residues were filtered using a 70-µm strainer.

The TILs were enriched using the Percoll density gradient precipitation method. First, Percoll (GE Healthcare, Cat. #17-0891-02) was diluted 9:1 with 10×PBS, and further diluted into 80% and 40% Percoll solutions with DMEM medium containing 10% FBS. The dissociated cells were collected (centrifugation at 500×g for 5 min) and resuspended in 40% Percoll. The suspension was carefully transferred into a 15-ml tube containing 80% Percoll to make the liquid stratified as much as possible. After centrifugation (500×g for 10 min), the intermediate suspended cells, majorly comprised of TILs (about 1 ml volume), were collected, washed and resuspended using ice-cold 1×PBS, and were prepared for further analysis.

## Measurement of serum or intratumour value of cytokines, proteins, alanine aminotransferase and aspartate aminotransferase

The intratumour or serum value of cytokines and proteins was determined using ELISA experiments according to the manufacturers' protocols (FastScan™ Total PD-L1 ELISA Kit Mouse–Preferred, Cell Signalling Technology, Cat. #41590; Mouse FLT3 ligand ELISA kit, Abcam, Cat. #ab275551; Mouse sCD40 ligand ELISA Kit, WESTANG, Cat. #F00411; Mouse TNF-a ELISA Kit, WESTANG, Cat. #F11630; Mouse IFN-γ ELISA Kit, Beyotime, Cat. # PI508; Mouse IL-12 p70 ELISA Kit, Beyotime, Cat. # PI530). To quantify the intratumour cytokines, 100 mg of tumour tissue was homogenised with 500 µl of ice-cold 1× PBS. The mixture was centrifuged (12,000×g for 5 min), and the supernatant was collected for further ELISA experiments. To quantify the serum cytokines, 100 µl of blood was centrifuged (3000×g for 5 min), and the supernatant serum was collected for further ELISA experiments. Serum samples were sent to BIOS Biological Company Shanghai for quantification of serum ALT and AST values.

## Flow cytometry experiments and data analysis

The immune cells derived from tumours, TdLNs, spleen and blood were prepared according to standard protocols. The Hilar lymph nodes, peripancreatic lymph nodes and superior mesentery lymph nodes were collected from each mouse for subsequent experiments. For cytoplasmic staining of the secreting proteins, the freshly isolated immune cells were incubated in RPMI-1640 Medium (Sigma, Cat. #RNBK0102) containing 10% FBS and 1×Protein Transport Inhibitor Cocktail (Invitrogen, Cat. #00-4980-93) for 8 h in the cell incubator. The Fixable Viability Stain 510 (BD Biosciences, Cat. #564406) were used to discriminate the viable or non-viable cells in the flow cytometry experiments. Next, the total cells were washed using 1×PBS containing 1% BSA, and blocked using CD16/32 antibody (BioLegend, Clone:93, Cat. #101320) for 30 min at room temperature. The cells for cytoplasmic or nuclear staining of IFN-γ, TNF-α, Ki67, and FoxP3 were further processed using intracellular fixation & permeabilization sets (BD, Cat. #554714 or Invitrogen, Cat. #225870) according to the manufacturer's protocols. Finally, the cells were incubated in 1×PBS (containing 1% BSA) or permeabilization buffer containing fluorescent-labelled flow cytometric antibodies, shaking at 4 °C for 12 h. The following staining protocols were designed to present: (1) overall distribution of immune cell, (2) T-cell function, (3) identification of DC cell subsets, (4) identification of DC cell function, (5) identification the M1/M2 phenotype of tumour-infiltrating macrophages and identification of co-stimulatory molecules, respectively. Panel-1: Fixable Viability Stain 510 (1:1000 dilution, BD Biosciences, Cat. #564406, RRID: AB_2869572), APC-eFluor™780 Rat anti-mouse CD45 Antibody (1:50 dilution, eBioscience, Cat. #47-0451-82, Clone: 30-F11, RRID: AB_1548781), FITC Rat anti-mouse CD3 Antibody (1:50 dilution, eBioscience, Cat. #11-0032-82, Clone: 17A2, RRID: AB_2572431), PE-Cyanine7 Rat anti-mouse CD8A Antibody (1:50 dilution, eBioscience, Cat. #25-0081-81, Clone: 53-6.7, RRID: AB_469583), BV421 Rat anti-Mouse CD4 Antibody (1:50 dilution, BD Biosciences, Cat. #562891, Clone: GK1.5, RRID:AB_2737870), PE-eFluor™ 610 Rat anti-mouse CD11B Antibody (1:50 dilution, eBioscience, Cat. #61-0112-80, Clone: M1/70, RRID: AB_2574527), PE Rat anti-mouse Ly-6G Antibody (1:50 dilution, BD Biosciences, Cat. #551461, Clone: 1A8, RRID: AB_394208), APC Mouse anti-mouse NK-1.1 Antibody (1:50 dilution, BD Biosciences, Cat. #550627, Clone: PK136, RRID: AB_398463), BUV395 Rat anti-mouse CD273 Antibody (1: 50 dilution, BD Biosciences, Cat. #565102, Clone: TY25, RRID: AB_2739068), Brilliant Violet 711™ Rat anti-mouse CD274 (B7-H1, PD-L1) Antibody (1:50 dilution, BioLegend, Cat. #124319, Clone: 10 F.9G2, RRID: AB_2563619). Panel-2: Fixable Viability Stain 510 (1:1000 dilution, BD Biosciences, Cat. #564406, RRID: AB_2869572), APC-eFluor™780 Rat anti-mouse CD45 Antibody (1:50 dilution, eBioscience, Cat. #47-0451-82, Clone: 30-F11, RRID: AB_1548781), FITC Rat anti-mouse CD3 Antibody (1:50 dilution, eBioscience, Cat. #11-0032-82, Clone: 17A2, RRID: AB_2572431), PE-Cyanine7 Rat anti-mouse CD8A Antibody (1:50 dilution, eBioscience, Cat. #25-0081-81, Clone: 53-6.7, RRID: AB_469583), BV421 Rat anti-Mouse CD4 Antibody (1:50 dilution, BD Biosciences, Cat. #562891, Clone: GK1.5, RRID: AB_2737870), PE Rat anti-mouse CD25 Antibody (1:100 dilution, eBioscience, Cat. #12-0251-82, Clone: PC61.5, RRID: AB_465607) or PE Mouse anti-human/mouse Granzyme B Recombinant Antibody (1:50 dilution, BioLegend, Cat. #372207, Clone: QA16A02, RRID: AB_2687031), APC Rat anti-human/mouse FOXP3 Antibody (1:50 dilution, eBioscience, Cat. #17-5773-80, Clone: FJK-16s, RRID: AB_469456), V450 Mouse anti-mouse/human Ki67 Antibody (1:50 dilution, BD Biosciences, Cat. #561281, Clone: B56, RRID: AB_10613816), Brilliant Violet 711™ Rat anti-mouse TNF-α Antibody (1:50 dilution, BioLegend, Cat. #506349, Clone: MP6-XT22, RRID: AB_2629800), BUV737 Rat Anti-Mouse IFN-γ Antibody (1:50 dilution, BD Biosciences, Cat. #612769, Clone: XMG1.2). Panel-3: Fixable Viability Stain 510 (1:1000 dilution, BD Biosciences, Cat. #564406, RRID: AB_2869572), APC-eFluor™780 Rat anti-mouse CD45 Antibody (1:50 dilution, eBioscience,

Cat. #47-0451-82, Clone: 30-F11, RRID: AB_1548781), FITC Rat anti-mouse CD3 Antibody (1:50 dilution, eBioscience, Cat. #11-0032-82, Clone: 17A2, RRID: AB_2572431), PE-Cyanine7 Rat anti-mouse CD8A Antibody (1:50 dilution, eBioscience, Cat. #25-0081-81, Clone: 53-6.7, RRID: AB_469583), BV421 Rat anti-Mouse CD4 Antibody (1:50 dilution, BD Biosciences, Cat. #562891, Clone: GK1.5, RRID: AB_2737870), PE Armenian Hamster anti-mouse CD152 Antibody (1:50 dilution, BioLegend, Cat. #106305, Clone: UC10-4B9, RRID: AB_313254), APC Mouse anti-mouse NK-1.1 Antibody (1:50 dilution, BD Biosciences, Cat. #550627, Clone:PK136, RRID: AB_398463), PerCP/Cyanine5.5 Rat anti-mouse CD223 (LAG-3) Antibody (1:50 dilution, BioLegend, Cat. #125212, Clone: C9B7W, RRID: AB_2561517), Brilliant Violet 711™ Rat anti-mouse CD366 (Tim-3) Antibody (1:50 dilution, BioLegend, Cat. #119727, Clone: RMT3-23, RRID: AB_2716208), BUV395 Rat anti-Mouse CD273 Antibody (1:50 dilution, BD Biosciences, Cat. #565102, Clone: TY25, RRID: AB_2739068). Panel-4.1: Fixable Viability Stain 510 (1:1000 dilution, BD Biosciences, Cat. #564406, RRID: AB_2869572), APC-eFluor™780 Rat anti-mouse CD45 Antibody (1:50 dilution, eBioscience, Cat. #47-0451-82, Clone: 30-F11, RRID: AB_1548781), FITC Rat anti-mouse I-A/I-E Antibody (1:50 dilution, BioLegend, Cat. #107606, Clone: M5/114.15.2, RRID: AB_313321), PE/Cyanine7 Mouse anti-mouse CD64 (FcγRI) Antibody (1:50 dilution, BioLegend, Cat. #139314, Clone: X54-5/7.1, RRID:AB_2563904), Brilliant Violet 421™ Rat Anti-mouse CD24 Antibody (1:50 dilution, BioLegend, Cat. #101826, Clone: M1/69, RRID: AB_2563508), PE Armenian Hamster anti-mouse CD11c Antibody (1:50 dilution, BioLegend, Cat. #117308, Clone: N418, RRID: AB_313777), PE-eFluor™ 610 anti-mouse CD11B Antibody (1:50 dilution, eBioscience, Cat. #61-0112-80, Clone: M1/70, RRID: AB_2574527), APC Mouse anti-mouse H-2Kb bound to SIINFEKL Antibody (1:50 dilution, BioLegend, Cat. #141605, Clone: 25-D1.16, RRID: AB_11219402), BUV395 Rat anti-mouse CD103 Antibody (1:50 dilution, BD Biosciences, Cat. #740238, Clone: M290, RRID:AB_2739985), Brilliant Violet 711™ Rat anti-mouse CD274 (B7-H1, PD-L1) Antibody (1:50 dilution, BioLegend, Cat. #124319, Clone: 10 F.9G2, RRID: AB_2563619). Panel-4.2: Fixable Viability Stain 510 (1:1000 dilution, BD Biosciences, Cat. #564406, RRID: AB_2869572), APC Mouse anti-mouse H-2Kd Antibody (1:50 dilution, BioLegend, Cat. #116620, Clone: SF1-1.1, RRID: AB_10645328) or anti-mouse H-2Kb Antibody (1:50 dilution, BioLedend, Cat. #116518, Clone: AF6-88.5, RRID: AB_10564404), FITC Rat anti-mouse I-A/I-E Antibody (1:50 dilution, BioLegend, Cat. #107606, Clone: M5/114.15.2, RRID: AB_313321), PE/Cyanine7 Mouse anti-mouse CD64 (FcγRI) Antibody (1:50 dilution, BioLegend, Cat. #139314, Clone: X54-5/7.1, RRID: AB_2563904), Brilliant Violet 421™ Rat anti-mouse CD24 Antibody (1:50 dilution, BioLegend, Cat. #101826, Clone: M1/69, RRID: AB_2563508), PE Armenian Hamster anti-mouse CD11c Antibody (1:50 dilution, BioLegend, Cat. #117308, Clone: N418, RRID: AB_313777), PE-eFluor™ 610 anti-mouse CD11B Antibody (1:50 dilution, eBioscience, Cat. #61-0112-80, Clone: M1/70, RRID: AB_2574527), APC Anti-mouse CD86 Antibody (1:50 dilution, BioLegend, Cat. #105012, Clone: GL-1, RRID: AB_493342), BUV395 Rat anti-Mouse CD103 (1:50 dilution, BD Biosciences, Cat. #740238, Clone:M290, RRID: AB_2739985), Brilliant Violet 711™ Dog anti-mouse CD80 Antibody (BioLegend, Cat. #104743, Clone: 16-10A1, RRID: AB_2810338), BUV737 Rat anti-Mouse CD40 Antibody (BD Biosciences, Cat. #741749, Clone:3/23, RRID: AB_2871115). Panel for identification the M1/M2 macrophage phenotypes: Fixable Viability Stain 510 (1:1000 dilution, BD Biosciences, Cat. #564406, RRID: AB_2869572), APC-eFluor™780 Rat anti-mouse CD45 Antibody (1:50 dilution, eBioscience, Cat. #47-0451-82, Clone: 30-F11, RRID: AB_1548781) ; FITC Armenian Hamster anti-mouse CD80 Antibody (1:50 dilution, BioLegend, Cat. #104706, Clone: 16-10A1, RRID: AB_313127) ; PE/Cyanine7 Rat anti-mouse CD86 Antibody (1:50 dilution, BioLegend, Cat. #105014, Clone: GL-1, RRID: AB_439783) ; Brilliant Violet 421™ Rat anti-mouse CD206 (MMR) Antibody (1:50 dilution, BioLegend, Cat. #141717, Clone: C068C2, RRID: AB_2562232); PE Rat anti-mouse CD163 Antibody (1:50 dilution, BioLegend, Cat. #155308,

Clone: S15049I, RRID:AB_2814062); PerCP-Cy™5.5 Rat anti-mouse CD11b Antibody (1:50 dilution, eBioscience, Cat. #61-0112-80, Clone: M1/70, RRID: AB_2574527); BUV395 Rat Anti-Mouse F4/80 (1:50 dilution, BD Biosciences, Cat. #565614, Clone: T45-2342, RRID: AB_2739304). The experiments were performed using BD LSR Fortessa X-20 instrument, and the data were analysed using FlowJo software (version 10.7.1).

## Immunohistochemistry and immunofluorescence staining

Tissues were harvested and fixed in 10% buffered formalin phosphate for over 48 h. The fixed tissues were embedded in paraffin, and sliced into sequential 8-μm thick slides. The H&E, Alcian-Blue, and Sirius-Red staining were performed according to a standard protocol. The TUNEL staining was performed using the TUNEL staining kit (Beyotime, Cat. #C1086) according to the manufacturer's protocol. The immunohistochemistry and immunofluorescence staining were performed using the paraffin-embedded slides according to a standard protocol. The Tyramide SuperBoost Kits (Invitrogen, Cat. #B40912-40926) were used for multicolour immunofluorescence staining. The NanoZoomer S360 (Hamamatsu Photonics, Hamamatsu, Japan) was used to scan slides of IHC staining, and the Pannoramic MIDI was used to scan slides of IF staining (3DHISTECH Digital Pathology Company, Budapest, Hungary). The anti-CD45 antibody (1:2000 dilution for IHC and 1:50 dilution for IF, ProteinTech, Cat. #60287-1-Ig, Clone:4E9B2, RRID: AB_2881404), anti-FoxP3 (1:200 dilution for IHC and 1:100 dilution for IF, Cell Signaling Technology, Cat. #12653, Clone: D6O8R, RRID: AB_2797979), anti-CD8 antibody (1:2000 dilution for IHC, Abcam, Cat. #ab209775, Clone: EPR20305, RRID: AB_2860566), anti-CD4 antibody (1:1000 dilution for IHC, Abcam, Cat. #ab183685, Clone: EPR19514, RRID: AB_2686917), anti-CD103 antibody (1:1000 dilution for IHC, Abcam, Cat. #ab224202, Clone: EPR22590-27, RRID: AB_2936238), anti-CK19 antibody(1:3000 dilution for IHC and 1:100 dilution for IF, ProteinTech, Cat. #10712-1-AP, RRID: AB_2133325) and anti-CD31 antibody (1:2000 dilution for IHC, Abcam, Cat. #ab182981, Clone: EPR17259, RRID: AB_2920881) were used for IHC- or IF- staining. We thank the BIOS Biological Company Shanghai for their assistance in tissue processing and image scanning.

## Western blotting

To perform western blotting, proteins were extracted from tumour tissues that had been treated with RIPA lysis buffer using a Nuclear and Cytoplasmic Protein Extraction Kit (Beyotime, Cat. #P0028). A Bradford reagent (Sigma, Cat. # B6916) was used to determine the protein concentration. The western blot analysis was performed in accordance with industry standards. Total RNA was extracted from the tissue using Trizol reagent (Invitrogen, Cat. #15596018). The anti-CK19 antibody (1:2000 dilution, ProteinTech, Cat. #60187-1-Ig, Clone: 3G1E4, RRID: AB_10859834), anti-TNF-α antibody (1:1000 dilution, ProteinTech, Cat. #60291-1-Ig, Clone: 7B8A11, RRID: AB_2833255), anti-IFN-γ antibody (1:1000 dilution, Invitrogen, Cat. #PA1-24782, RRID: AB_794536), anti-HMGB1 antibody (1:1000 dilution, Abcam, Cat. #ab18256, RRID:AB_444360), anti-SpCas9 antibody (1:10000 dilution, Abcam, Cat. #ab189380, Clone: EPR18991, RRID:), anti-β-Actin antibody (1:1000 dilution, CST, Cat. #3700, Clone: 8H10D10, RRID: AB_2242334), anti-GAPDH antibody (1:5000 dilution, Abcam, Cat. #ab9484, Clone: mAbcam9484, RRID: RRID:AB_307274), and anti-PD-L1 antibody (1:2000 dilution, ProteinTech, Cat. #66248-1-Ig, Clone: 2B11D11, RRID: AB_2756526) were used for western blotting.

## RNA-sequencing and data analysis

The tumour samples for transcriptome profiling were collected and kept in liquid nitrogen. Total RNA extraction, RNA quantification, and library preparation were performed by Novogene (Tianjin, China). The Illumina HiSeqX10 was used for high-throughput sequencing. Approximately 6 Gbase raw data was obtained for each sample. The

fastp software (version 0.20.1) was used for quality control and adaptor removal of the raw paired-end reads. The Bowtie2 software was used to map the qualified reads to the mm10 genome. The Htseq-count software was used to quantify the gene expression counts, and the DEseq2 package (version 3.12, R version 3.6.3) was used to identify the differentially expressed genes between IRE-treated or non-IRE-treated tumours. The R BioMart package (version 0.7) was used to convert the murine gene IDs to the corresponding human gene IDs, and then GSEApy software (version 0.9.18) was used for pre-ranked GSEA analysis.

## Analysis of the TCGA datasets

The transcriptome profiles (Htseq-counts) and clinical data of the TCGA-PAAD cohort were obtained using the R TCGAbiolinks package (version 2.20.1). Briefly, the reads-count data was converted to TPM (Transcripts Per Kilobase of exon model per Million mapped reads) using custom drafted script. The gene expression features of 24 kinds of tumour-infiltrating immune cells were set according to a previous report[59]. The R GSVA package (version 1.40.1) was used for the single-sample gene set enrichment analysis to identify the infiltration feature of immune cells. The R ConsensusClusterPlus Package (version 1.58.0) and factoextra package (version 1.0.7) were used to identify the unsupervised clustering of the immune-infiltrating data.

## Statistical analysis and reproducibility

Statistical analysis was performed using R (version 3.6.3) and Graph-Pad Prism 8 software (San Diego, CA, USA). The initial sample size was estimated using the PASS software (Version 2015). Sample size or biological replicate was presented in the figure legends. For each result shown by representative images, the stability of the results has been verified by at least three independent experiments. The Pearson analysis was performed to identify the correlation between two continuous variables. Survival differences between different groups were compared using the two-sided log-rank test. Dunnett $t$ test and unpaired Student's $t$ test was used to compare continuous variables between the experimental group and multiple control groups. Two-sided $P$ value less than 0.05 was considered statistically significant.

## Reporting summary

Further information on research design is available in the Nature Portfolio Reporting Summary linked to this article.

## Data availability

The raw sequencing data generated in this study have been deposited in the NCBI Gene Expression Omnibus (GEO) database under accession code GSE215417. The data and detailed information of the experiments are fully publicly available. The remaining data generated in this study are provided in the Supplementary Information and Source Data file. Source data are provided with this paper.

## Code availability

In the analysis of TCGA datasets, reads-count data was converted to TPM using custom drafted script. The reads-count to TPM convert script are fully publicly available (https://github.com/sdtaliuxiaoyu/read-counts-to-TPM.git).

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

## Acknowledgements

This study was founded by the National Key Research and Development Program of China (2020YFA0908200 to W.C.), National Natural Science Foundation of China (No. 82272089 to Z.M.W.; 81930051 to W.C.), Clinical Key Specialist Construction Project of Shanghai Municipal Health Commission (Interventional Radiology, [No.shslczdzk06002] and 3D Printing [No. Shslczdzk07002] to Z.M.W.), Shanghai Key Specialty Construction Project (No. ZK2019A02 to Z.M.W.), and Guang Ci Professorship Program of Ruijin Hospital Shanghai Jiao Tong University School of Medicine (to W.C.). We thank Dr. Lin Qiu (Shanghai Institute of Nutrition and Health, Chinese Academy of Sciences, Shanghai, China) for her guidance on flow cytometry experiments. We thank Prof. Yuezhen Deng (Xiangya Cancer Centre, Xiangya Hospital, Central South University, Changsha, China) and Dr. Tao Wei (Department of General Surgery, The First Affiliated Hospital, Zhejiang University School of Medicine, Hangzhou, China) for their kind help and suggestion on molecular biology and biochemistry experiments.

## Author contributions

W.C., Z.M.W., Z.Q. and X.L. designed the project. Y.P.Z. synthesised and characterised the biomaterials. X.L. and Y.C. established the IRE ablation mouse model. X.L., Y.C., W.H., Q.S., Z.Z.W. and Y.F.Z. performed the animal experiment. X.L., Y.C., Z.Z.W. and Z.Y.W. performed the cell experiment. X.L., Y.C. and Y.F.Z. performed the biochemical experiments. X.L., Y.C. and Z.Z.W. performed the flow cytometry experiment and analysed the data. X.L. and Y.P.Z. designed and performed the bioinformatic and statistical analysis.

W.C., Z.M.W. and X.L. interpreted the data and drafted the manuscript. X.D., Z.Y.W. and Z.Q. participated in manuscript discussions and provided valuable suggestions. All authors took part in the discussion and revision of the manuscript. We declare that all authors contribute to the study and carry accountability, and authors fulfil the author criteria.

## Competing interests

The authors declare no competing interests.
