## [Peer Review File · Nature Communications]

Interventional hydrogel microsphere vaccine as an immune amplifier for activated antitumour immunity after ablation therapyEditorial Note: Parts of this Peer Review File have been redacted as indicated to maintain the confidentiality of comments which were intended for the editor only.

REVIEWER COMMENTS

Reviewer #1 (Remarks to the Author):

In this manuscript, the authors developed a hydrogel microsphere vaccine as an immune amplifier for post-ablation cancer immunotherapy. The hydrogel microsphere vaccine acts as a general immune amplifier to trigger a rocket-like amplification of the cDC1-mediated antigen cross-presentation cascade, resulting in dramatic amplification of the anti-tumour immunity in pancreatic cancer.

The study provides evidence on methodological and technological advancement of dual-cytokine preparation for efficient antitumor therapy for solid tumors. The new approach addressed safety concerns associated with potential off-target and pan-immune remodulation, due to its localized delivery and prolonged retention in the tumors.

This approach also has strong translational potential. If such hydrogel vaccine system would be GMP produced, it could be swiftly tested in clinical trials. Particularly since similar approaches, such as CD40 agonists and FLT3L, have been shown to be effective in preclinical models. This study also highlights the synergistic effect of the hydrogel microsphere vaccine with ablation therapy and antigen release to promote endogenous antitumor immunity and produce an abscopal effect.

Overall, this study is of great novelty and integrity, and it offers a new immunotherapeutic modality for solid tumors. Therefore, I support acceptance of this work after addressing the following issues.

Major issues:

1. The molecular features of the KPC cells should be thoroughly characterized in vitro. This includes checking the expression of pancreatic cancer markers and immune checkpoints.
2. In Figure S15, the authors declared that sporadic inflammatory responses were observed in the lungs of mice receiving an i.p injection of α CD40 and FLT3L cytokines according to H&E staining. IHC or IF analysis could be added to confirm the organ toxicity.
3. Activating immune responses in pancreatic cancer via small molecular drugs and CD40 agonism has just been reported (Jiang et al., *Gastroenterology*, 2022). What is the difference and advantage of the hydrogel microsphere vaccine system? The comparison and discussion should be extended in this direction.

Minor issues:

1. Please check the schematic figures and remove the watermark traces left by the drawing software.
2. In Figure 6a, the definition of "ISIM" needs to be clarified.
3. In Figure 6d (right panel), the order of the treatment groups should be mislabeled.
4. Please uniform the format of FLT3L (FLT3L or Flt3L in text and different figures).
5. The GSE number for the RNA-sequencing data in the Data availability section was not provided.
6. In Figure Legends 3, "CD8+ T" should be "CD8+ T". Some other typo errors should also be carefully checked.

Reviewer #2 (Remarks to the Author):

This article reports the potential of hydrogel microspheres for enhancing antitumor immunity after tumor ablation. The authors propose that immune amplification function of hyaluronic acid-based hydrogel microspheres could play a major role in boosting antitumor immunity. I evaluate that the manuscript does not meet the standard of Nature Comm. My comments are as follows.

1. In this study, the hydrogel microsphere system has several elements, Cas9-PDL1 plasmid DNA, cationic liposome, CD40L cytokine, CaCO₃ nanoparticle, FLT3L, and hyaluronic acid derivatives (HAMA). Combination of several components raises up a question on what is the innovative point of this study.

2. In Figure 1, the authors proposed the controlled release of CD40L cytokine from hyaluronic acid-based hydrogel in acidid tumor microenvironment. The authors should show the pH dependence of CD40L release from the hydrogel.
3. What is the driving force of calcium ion loading in polymeric micelles?
4. The authors should provide more extensive physicochemical characterization data of hydrogel microspheres. For example, the size, and zeta potential of cationic liposomes before and after Cas9 plasmid entrapment. The authors should provide the data on the amount of Cas9 plasmid DNA entrapped in the cationic liposome.
5. What is the rationale by which hyaluronic acid-based hydrogel show pH-dependent release pattern? Whis is the pH-sensitive moieties in the hydrogel matrix?
6. The authors should show evidence on the actual acidity values of residual tumors after tumor ablation.
7. The authors should provide the cell populations of tumor microenvironments before and after the hydrogel microsphere vaccination.
8. To support the immunotherapeutic function of hydrogel microsphere vaccines, the specificity against tumor cell type is essential. The authors should challenge the mice with distant tumors, which are different from primary tumor cell types.
9. The authors need to provide in vitro and in vivo evidence of gene silencing of PD-L1 protein by Cas9-PD-L1 encoding plamid DNA.
10. The authors should provide the intracellular expression of Cas9 protein in the tumor cells and in vivo tumor tissues.
11. The authors focused on the recruitment and amplification of cDC1s by FLT3L. How did the authors differentiate the recruited cDC1s from amplified cCD1s?
12. The authors should test which immune cells in tumor microenvironment were the major cell types for preventing the growth of distant tumors.

Minor

1. In line 489, "The plasmid-to-lip ratio is 1:1, 1:1.5,1:2,1:2.5, and 1:3 according to the experimental design". The authors should clarify the unit of plasmid DNA to lipid ratios. Is it N/P ratio or weight ratio?
2. The authors should describe which plasmid-to liposome ratio was used for the final hydrogel system.
3. In method section, what is the lipid composition of cationic liposome?

Reviewer #3 (Remarks to the Author):

In this manuscript Liu et al. develop and study the effects of a hydrogel microsphere vaccine in the treatment of pancreatic cancer. The authors report that treatment of orthotopic KPC pancreatic mouse models with the hydrogel microsphere vaccine increased animal survival and inhibited the growth of distant metastasis. This effect is due to an activation of cDC1 cells and consequent antigen presentation and activation of CD8 Tcells.

Overall, the manuscript is well written, well positioned in the state-of-the-art of the field and the authors provide clear assays that support their claims making this manuscript a good addition to the literature. However, the data presented does not fully support the conclusions and some issues need to be addressed to clarify the results:

- 1) Regarding the data in figure 2p and 2q, the authors should check if the hydrogel microsphere vaccine had any effects on cDC1s' proliferation and differentiation.
- 2) On figure 3k and 4b the authors show IHCs for CD45 on mice pancreatic tumors to assess whether IRES ablation and the hydrogel microsphere vaccine affected immune cell infiltration in orthotopic pancreatic tumors. Because the study focuses on cDC1s and CD8+ T cells, the authors should also perform staining for these immune cell types. This would allow the assessment of whether IRES ablation and the hydrogel microsphere vaccine being tested are able to promote the recruitment of these specific immune cells into the TME.

3) On line 289, while describing figure 4c, I believe the authors mistakenly wrote FLT3L, as in the figure it is the CD40L's highest concentration that lasted from 48 hours to 96 hours.

4) Figure 4b, the hydrogel microsphere vaccine group is labelled as "IRE+Hydrogel microspheres vaccine", while on figures 4e, 4f, 4k and 4l it is labelled as "Group 5". Please change these labels to something more uniform. Also, I believe calling it Group 5 might cause confusion as all the other groups are controls.

5) Please explain, in the results section, the treatment groups presented on figures 4i, 4j, 4m and figures 5a-5e. Are these the same as Groups 1 to 4 presented in other figures? This should be clarified and the names given should be simplified in a way that the reader immediately knows to which group the authors are referring to while describing the results.

6) In Materials and Methods the authors say that tumor volume was evaluated every 3 days but the method used is not mentioned. Also, on the experiment depicted on figure 6, did the authors assess tumor volume and weight at the time of euthanasia? Was there any difference in tumor volume between groups?

7) In Materials and Methods it is not mentioned how the metastases were evaluated. Also, the authors mention on line 406 that the metastases that are being evaluated are "distant subcutaneous metastases". Orthotopic pancreatic mouse models don't develop subcutaneous metastases, but rather liver and lung, and sometimes peritoneal dissemination can also be seen. Could you please clarify this issue?

8) The authors never mention some important immune cells that are crucial in the TME and inflammation response, like macrophages. Analysis for these innate immune cells and, additionally, if these cells change fate - from pro to antitumorigenic, or vice-versa, especially in the survival studies, should be included.

Minor points:

Figure S7 misses legend for panels c and d.

Figure 5k can be deleted. I believe it is a repetition of 1d.

Response letter

Point-to-point response to reviewers

Reviewer #1

In this manuscript, the authors developed a hydrogel microsphere vaccine as an immune amplifier for post-ablation cancer immunotherapy. The hydrogel microsphere vaccine acts as a general immune amplifier to trigger a rocket-like amplification of the cDC1-mediated antigen cross-presentation cascade, resulting in dramatic amplification of the anti-tumour immunity in pancreatic cancer.

The study provides evidence on methodological and technological advancement of dual-cytokine preparation for efficient antitumor therapy for solid tumors. The new approach addressed safety concerns associated with potential off-target and pan-immune remodulation, due to its localized delivery and prolonged retention in the tumors.

This approach also has strong translational potential. If such hydrogel vaccine system would be GMP produced, it could be swiftly tested in clinical trials. Particularly since similar approaches, such as CD40 agonists and FLT3L, have been shown to be effective in preclinical models. This study also highlights the synergistic effect of the hydrogel microsphere vaccine with ablation therapy and antigen release to promote endogenous antitumor immunity and produce an abscopal effect. Overall, this study is of great novelty and integrity, and it offers a new immunotherapeutic modality for solid tumors. Therefore, I support acceptance of this work after addressing the following

issues.

Major issues

Comment 1. The molecular features of the KPC cells should be thoroughly characterized in vitro. This includes checking the expression of pancreatic cancer markers and immune checkpoints.

Response: Thanks for the suggestion. We have characterized the expression level of major molecular markers of KPC and Panc02 cells using Western Blot assay. The results were presented in **Supplementary Fig.23a**.

Supplementary Fig.23a. Western Blot assay demonstrating the expression level of HMGB1, CK19 and CD274 (PD-L1) in KPC and Panc02 cells.

Comment 2. In Figure S15, the authors declared that sporadic inflammatory responses were observed in the lungs of mice receiving an i.p injection of α CD40 and FLT3L cytokines according to H&E staining. IHC or IF analysis could be added to confirm the organ toxicity.

Response : Thanks for the valuable suggestion. We have taken the reviewer's suggestions and supplemented CD45 IHC-staining to further confirm the organ toxicity. The results suggested that local injection of the hydrogel microsphere vaccine did not

induce significant inflammatory responses in heart, kidney, liver, and lung. The results were updated in **Supplementary Fig. 19**.

Supplementary Fig. 19. Representative H&E and CD45 immunohistochemical staining images of major organs in KPC orthotopic tumor bearing mice receiving different therapies. No obvious signs of organ damage appeared in the hydrogel microspheres treated mice. Sporadic inflammatory responses were observed in the lungs of mice receiving i.p. injection of CD40 agonist and FLT3L cytokines.

Comment 3. Activating immune responses in pancreatic cancer via small molecular drugs and CD40 agonism has just been reported (Jiang et al., Gastroenterology, 2022). What is the difference and advantage of the hydrogel microsphere vaccine system? The comparison and discussion should be extended in this direction.

Response: We thank the reviewer for the valuable comment. We have carefully read the articles published by Jiang et al. [1], compared our work with theirs, and strengthened our novelties and advantages. Jiang and colleagues reported that MEK and autophagy

co-inhibition coupled with α CD40 invoked immune repolarization and was an attractive therapeutic approach for pancreatic cancer immunotherapy development. In their report, MEK and autophagy co-inhibition triggered the release of inflammatory cytokines in cancer cells, and these signals affected macrophages' polarization to favor an M1-like, antigen-presenting phenotype.

Compared with MEK and autophagy co-inhibition, irreversible electroporation ablation is an emerging local therapy for cancers. This non-thermal ablation therapy directly induces cancer cell apoptosis and triggers a strong inflammatory response. In our study, we highlighted the amplification of the CD8⁺ T-cell immune cascade induced by the activation of cDC1s. Compared with M1-like macrophages, cDC1 is currently recognized as the strongest antigen-presenting cell targeting CD8⁺ T cells, and this immune cascade amplification has superior efficiency and specificity [2,3].

In addition, Jiang and colleagues reported that the therapy mostly increased the CD301b⁻ subset of cDC2, whereas the CD301b⁺ subset was decreased in the conditions where α CD40 was used. More importantly, the moderate loss of cDC1s in the tumor microenvironment may impair antigen presentation to CD8⁺ T cells and attenuate the expansion and activation of CD8⁺ T cells. In our study, we highlighted that the combination of local and sequential administration of FLT3L and α CD40 could significantly enhance the cDC1-mediated antigen-presentation cascade. As we mentioned in the *Introduction* and *Discussion* sections, since myeloid cells are highly plastic, different combinations of cytokines and delivery strategies may lead to differences in the differentiation and activation status of myeloid cells. Here, we have

addressed the critical issue of efficient local amplification of tumor-resident cDC1s. Based on the hydrogel vaccine system, we proposed a novel local therapeutic strategy for pancreatic cancer, and elucidated the immune basis of this anti-tumor immune amplification strategy from three levels: local antitumor immune remodeling, draining lymph node immune response, and systemic adaptive antitumor immunity. Recent evidence has also suggested that the irreversible electroporation may induce immune activation and epitope expansion effects in prostate tumors [4]. The amplification of cDC1-mediated antigen-presentation cascade using hydrogel microsphere vaccine undoubtedly promoted the translational application of irreversible-electroporation based local therapies. In the revised manuscript, we have also reinforced our advantages and innovations with this previous study (*Discussion Section, Page 15, line 21-27*). We sincerely hope sincerely this will meet your criteria.

References

- [1] Jiang Honglin, Courau Tristan, Borison Joseph et al. Activating Immune Recognition in Pancreatic Ductal Adenocarcinoma via Autophagy Inhibition, MEK Blockade, and CD40 Agonism.[J]. *Gastroenterology*, 2022, 162: 590-603.e14.
- [2] Giampazolias Evangelos, Schulz Oliver, Lim Kok Haw Jonathan et al. Secreted gelsolin inhibits DNGR-1-dependent cross-presentation and cancer immunity.[J]. *Cell*, 2021, 184: 4016-4031.e22.
- [3] Schenkel Jason M, Herbst Rebecca H, Canner David et al. Conventional type I dendritic cells maintain a reservoir of proliferative tumor-antigen specific TCF-1 CD8 T cells in tumor-draining lymph nodes.[J]. *Immunity*, 2021, 54: 2338-2353.e6.
- [4] Burbach Brandon J, O'Flanagan Stephen D, Shao Qi et al. Irreversible electroporation augments checkpoint immunotherapy in prostate cancer and promotes tumor antigen-specific tissue-resident memory CD8+ T cells.[J]. *Nat Commun*, 2021, 12: 3862.

Minor issues

Comment 4. Please check the schematic figures and remove the watermark traces left by the drawing software.

Response: Thanks for the kind reminder. We sincerely apologize for our carelessness. We have checked all the figures and made sure they were of high quality.

Comment 5. In Figure 6a, the definition of “ISIM” needs to be clarified.

Response: Thanks for the kind reminder. Here, “ISIM” is the abbreviation for “in situ immune modulation”. The definition of "ISIM" has been added in the revised figure legends (*Figure Legends 6, Page 31, line 3-4*).

Comment 6. In Figure 6d (right panel), the order of the treatment groups should be mislabeled.

Response: Thanks for the comment. We have checked Figure 6d and reorganized the figure panels and annotations.

Comment 7. Please uniform the format of FLT3L (FLT3L or Flt3L in text and different figures).

Response: Thanks for the kind suggestion. In the manuscript, Flt3l represents the mouse gene name, and the capitalized FLT3L and Flt3L refers to the corresponding soluble protein. We have checked the manuscript to make sure the correct words were used.

Comment 8. The GSE number for the RNA-sequencing data in the Data availability section was not provided.

Response: Thanks for the comment. The RNA sequencing data has now been uploaded correctly to The Gene Expression Omnibus (GSE215417) (**Data availability section, Page 20, line 8**).

Comment 9. In Figure Legends 3, “CD8+ T” should be “CD8+ T”. Some other typo errors should also be carefully checked.

Response: Thanks for the kind reminder. We sincerely apologize for our carelessness. In the revised manuscript, we have checked the manuscript to avoid the typographical errors.

Reviewer #2

This article reports the potential of hydrogel microspheres for enhancing antitumor immunity after tumor ablation. The authors propose that immune amplification function of hyaluronic acid-based hydrogel microspheres could play a major role in boosting antitumor immunity. I evaluate that the manuscript does not meet the standard of Nature Comm. My comments are as follows.

Major issues

Comment 1. In this study, the hydrogel microsphere system has several elements: Cas9-PDL1 plasmid DNA, cationic liposome, CD40L cytokine, CaCO₃ nanoparticle, FLT3L, and hyaluronic acid derivatives (HAMA). Combination of several components raises a question on what the innovative point of this study is.

Response: We thank the reviewer for the valuable comment. We understand the concerns of the reviewer. Hydrogel microspheres, liposomes, and calcium carbonate nanoparticles have been widely used in the development of drug delivery systems because of their good bio-affinity and bio-responsiveness. We can understand that the reviewer raised this concern from the perspective of materials science. We accept that developing entirely new materials and drug delivery systems is very innovative work. Meanwhile, we believe that the utilization and modification of current biomaterial carriers for new biological or medical applications offers equivalently significant innovative and clinical translational value [1-3].

In this study, we developed a locally injectable "hydrogel microsphere

immunomodulatory chamber" using common and biocompatible materials to achieve programmatic release of FLT3L and CD40 agonist. One of the most significant characteristics of the vaccine is the precise amplification of the antitumor immune effect of cDC1/CD8⁺T cells. Within the context of immunotherapy, the strategy of targeted amplification of the immune cascade cDC1/CD8⁺T is highly innovative. The amplification and activation of CD8⁺ T cells are the core of current immunotherapy. The canonical immune checkpoint blockade (ICB) therapy relies on blocking immunosuppressive signaling, thus reversing CD8⁺ T cell exhaustion. Numerous preclinical studies have attempted to enhance ICB strategies to boost the antitumor immune effects. However, for solid tumors with a higher degree of malignancy, such as pancreatic cancer, glioma, and triple-negative breast cancer, the ICB strategies are hardly successful. Even among tumors responsive to ICB therapy, the overall response rate of the patients remains poor (20%-30%) [4]. This is primarily owing to the aberrant or weak immunological effects of these malignancies. Even though some studies expect to boost antitumor immunity by delivering antigen or triggering a tumor inflammatory response, how to assure adequate immune response and tumor targeting effect continues to be a major obstacle for the majority of the studies. As we mentioned in the introduction section, in recent years, cDC1-mediated antigen presentation and antitumor immunity have gained increasing attention. However, there is still no effective strategy to locally activate and augment cDC1-mediated anti-tumor immunity. Taking the advantages of biomaterials and interventional therapy, we successfully achieved the adequate amplification of the cDC1/CD8⁺ T immune cascade. This provides a novel

approach for achieving antitumor immune amplification in solid tumors and provides a new possibility for immunotherapy based on the amplification immune effects in solid tumors.

Next, we applied the immune amplification strategy to pancreatic cancer, a solid tumor with a high mortality rate and a very poor response to current immunotherapies. The hydrogel vaccine and IRE ablation demonstrated a high degree of responsiveness in advanced pancreatic cancer by attenuating local tumor progression and eliciting strong abscopal effects. We have clearly elucidated the mechanism by which cDC1 activates adaptive immune amplification of CD8⁺ T cells intratumor and within TdLNs, confirming the specificity and safety of this new therapy. Although we used a simple combination of materials, we designed the hydrogel vaccine according to the biological mechanism of the disease to achieve an antitumour immune cascade amplification, and finally achieve the purpose of efficient intratumoral immunotherapy for pancreatic cancer. Considering the current clinical predicament in the treatment of advanced pancreatic cancers, we believe this new strategy gains novelty and potential for clinical translation from the perspective of oncology.

Revisions have been made in the *Introduction (Page 3, line 25-28; Page 4, line 10-14; Page 5, line 1-4)* and *Discussion (Page 15, line 21-30)* sections to highlight the originality and novelty of the work. And we have also tried our best to solve the problems raised by reviewers to enhance the quality of our work. We thank the reviewer again for the valuable comment. We sincerely hope this explanation could clarify this issue.

References

- [1] Li Wen-Hao, Su Jing-Yun, Li Yan-Mei, Rational Design of T-Cell- and B-Cell-Based Therapeutic Cancer Vaccines. [J]. *Acc Chem Res*, 2022, 55: 2660-2671.
- [2] Zhang Jing, Chen Chen, Li Anning et al. Immunostimulant hydrogel for the inhibition of malignant glioma relapse post-resection. [J]. *Nat Nanotechnol*, 2021, 16: 538-548.
- [2] Majumder Poulami, Singh Anand, Wang Ziqiu et al. Surface-fill hydrogel attenuates the oncogenic signature of complex anatomical surface cancer in a single application. [J]. *Nat Nanotechnol*, 2021, 16: 1251-1259.
- [4] Huang Qizhao, Wu Xia, Wang Zhiming et al. The primordial differentiation of tumor-specific memory CD8 T cells as bona fide responders to PD-1/PD-L1 blockade in draining lymph nodes. [J]. *Cell*, 2022, doi: 10.1016/j.cell.2022.09.020, online publication.

Comment 2. In Figure 1, the authors proposed the controlled release of CD40L cytokine from hyaluronic acid-based hydrogel in an acidified tumor microenvironment. The authors should show the pH dependence of CD40L release from the hydrogel.

Response: Thanks for the valuable comment. In the revised manuscript, we simulated pH value of the tumor microenvironment and performed the release of CD40L and FLT3L from the hydrogel microspheres in $1 \times$ PBS solution at pH 6.8, which was shown in **Figure 2I**.

In this revision, the release of CD40L from the hydrogel microspheres system in $1 \times$ PBS at pH 7.4 and pH 6.8 was exhibited. Here, the pH value of 6.8 was chosen for two reasons. Firstly, the pH value of 6.8 have been widely accepted as the pH value for simulating the tumor microenvironment in vitro [1,2]. Secondly, according to

Comment 6, we have measured the true intratumour pH value before and after tumor ablation. The mean pH value of the tumor was around 6.8 before treatment, and the acidity was slightly increased after ablation, which ensured the release of CD40L. The soluble CD40L and FLT3L values were all detected by ELISA experiments.

Supplementary Figure 2a. The release of CD40L from CD40L@CaCO₃/FLT3L/Cas9@lip/HAMA hydrogel microsphere at pH value of 6.8 and 7.4 in 1×PBS solution system.

References:

- [1] Guo Qin, Li Xuwen, Zhou Wenxi et al. Sequentially Triggered Bacterial Outer Membrane Vesicles for Macrophage Metabolism Modulation and Tumor Metastasis Suppression. [J]. ACS Nano, 2021, doi:10.1021/acsnano.1c05613
- [2] Ying Zhao, Tianjiao Ji, Hai Wang et al., Self-assembled Peptide Nanoparticles as Tumor Microenvironment Activatable Probes for Tumor Imaging, J Controlled Release, 2014, 10, 177, 11-19.

Comment 3. What is the driving force of calcium ion loading in polymeric micelles?

Response : Thanks for the valuable comment. Here, we first synthesized an amphiphilic triblock polymer, PLGA-PEG-PLGA, and modified its end by carboxylation with succinic anhydride. The modified triblock polymer HOOC-PLGA-PEG-PLGA-COOH can be self-assembled into micelles in aqueous solution. The micelles include a halo of hydrophilic PEG and a core of carboxylated PLGA. The carboxyl groups in the micelle core can complex with supersaturated Ca^{2+} in solution to provide nucleation sites. While Ca^{2+} concentrates around the nucleation sites to increase the local supersaturation while spontaneously forming aggregates. Under the action of Ca^{2+} bridging and hydrogen bonding, the aggregates rearrange and self-assemble to form the calcium carbonate nanoparticles [1,2].

References:

- [1] Gao Yun-Xiang, Yu Shu-Hong, Cong Huaiping et al. Block-copolymer-controlled growth of CaCO_3 microrings.[J] .J Phys Chem B, 2006, 110: 6432-6.
- [2] Chen Wei, Wang Guohao, Yung Bryant C et al. Long-Acting Release Formulation of Exendin-4 Based on Biomimetic Mineralization for Type 2 Diabetes Therapy.[J] .ACS Nano, 2017, 11: 5062-5069.

Comment 4. The authors should provide more extensive physicochemical characterization data of hydrogel microspheres. For example, the size and zeta potential of cationic liposomes before and after Cas9 plasmid entrapment. The authors should provide the data on the amount of Cas9 plasmid DNA entrapped in

the cationic liposome.

Response: We thank the reviewer for the valuable comment. The size and zeta potential of cationic liposomes before and after Cas9 plasmid entrapment have been presented in **Figure 2**. The Z-average diameter of cationic liposomes is 140 nm, and the zeta potential is 24.9 mV, and the particle size of liposomes does not change much after loading the Cas9 plasmid. The results showed that the Z-average diameter is 124 nm, and the zeta potential drops to 9.52 mV of cas9@lip.

We also provide more extensive physicochemical characterization data of hydrogel microspheres in **Supplementary Fig. 1c-h**. The particle size statistics of the hydrogel microspheres and the photographs of the hydrogel microspheres loaded with CaCO₃ nanoparticles under white light conditions are supplemented. Subsequently, the FITC-loaded liposomes were complexed with hydrogel microspheres, and the fluorescence photos also proved the successful complexation of liposomes and hydrogel microspheres.

The benzene ring structure of the bases on the Cas9 plasmid DNA chain has strong UV absorption in the ultraviolet region, and its absorption peak is found at 258 nm by UV-vis spectrophotometer detection. We first established a standard curve with a known concentration of Cas9 plasmid DNA, and then inferred that Cas9 plasmid DNA was entrapped in the cationic liposome. The standard curve was shown in **Supplementary Fig. 2b** and the linear correlation coefficient R² is 99.98 by linear fitting. Then, by adding methanol at a volume ratio of 1:10 to the Cas9@lip to break the emulsion, the absorbance of Cas9 plasmid DNA in the solution after breaking the emulsion was

detected by UV-Vis, and then the concentration of Cas9 plasmid DNA was determined according to the standard curve obtained above. The encapsulation efficiency of Cas9 plasmid DNA in cationic liposomes is 56.07%.

Supplementary Figure 1. The physicochemical characterization data of hydrogel microspheres (c-h). (c) Hydrodynamic diameter of the lip; (d) Hydrodynamic diameter and zeta potential of the Cas9@lip; (e) Fluorescence photograph of FITC@lip/HAMA hydrogel microspheres; (f) The particle size statistics of hydrogel microspheres by image J; (g) The photograph of CaCO₃@HAMA hydrogel microspheres; (h) The enlarged image of CaCO₃@HAMA hydrogel microspheres.

Supplementary Figure 2 b. The standard curve of Cas9 plasmid DNA according to UV absorption analysis.

Comment 5. What is the rationale by which hyaluronic acid-based hydrogels show pH-dependent release patterns? What are the pH-sensitive moieties in the hydrogel matrix?

Response: Thanks for the comment. In the hydrogel microsphere system, CD40L was first loaded into the calcium carbonate nanoparticles, and the calcium carbonate nanoparticles could be decomposed under acidic pH conditions to release the loaded CD40L. The specific preparation process was to first load CD40L into HOOC-PLGA-PEG-PLGA-COOH micelles, and then form calcium carbonate nanoparticles around the micelles through biomineralization. The release of CD40L required calcium carbonate erosion in the slightly acidic environment of tumors.

Comment 6. The authors should show evidence on the actual acidity values of residual tumors after tumor ablation.

Response: Thanks for the valuable suggestion. In the revised manuscript, we assessed the pH value of the tumor before and after the ablation therapy using a pH micro-electrode (CAT#pH-500c 400-600um diameter, UNISENSE company, Denmark). About 3~4 mm of the pH micro-electrode was punctured into the tumor carefully. The pH values were measured before ablation, 0h (immediately after ablation), 24h, 48h and 72h after the ablation therapy. The mean pH value for each group is 6.84, 6.81, 6.75, 6.70, and 6.67, respectively (**Supplementary Fig. 6**).

There are no significant differences in pH values before and immediately after the ablation therapy. The intratumour pH values were slightly but significantly reduced after 24, 48h, and 72h after the IRE ablation therapy. The slight decrease in pH value after ablation was compatible to the release of CD40L from the hydrogel vaccine. It was previously reported that the intratumour pH value was slightly decreased after ablation therapy [1, 2]. This decrease in pH may be due to the weakening of microvascular drainage, accumulation of metabolites, and the release of waste products caused by cell damage and death.

Supplementary Fig. 6. Measurement of the intratumour pH value. The intratumour pH value before and after the IRE therapy. There were three mice in each group, and each mouse was tested for three times.

Reference

- [1] Nikfarjam Mehrdad, Muralidharan Vijayaragavan, Christophi Christopher, Mechanisms of focal heat destruction of liver tumors. [J]. J Surg Res, 2005, 127: 208-23.
- [2] Song C W, Kang M S, Rhee J G et al. The effect of hyperthermia on vascular function, pH, and cell survival. [J]. Radiology, 1980, 137: 795-803.

Comment 7. The authors should provide the cell populations of tumor microenvironments before and after the hydrogel microsphere vaccination.

Response: Thanks for the suggestion. Comparison of the tumor microenvironment before and after the vaccination in the same mouse model is challenging because of the limitations of the surgical models and sampling methods. Therefore, it is feasible to set up control and compare the changes of immune microenvironment. We have analyzed

the effects of hydrogel vaccination on the tumor immune microenvironment in the context of IRE therapy (**Figure 4d-j; Figure 5a-f; Supplementary Fig. 16b**). In these experiments, the blank CaCO₃ and hydrogel are used as controls. And hydrogels loading with different compositions are also used as controls for the hydrogel vaccine. The results showed that the hydrogel vaccine, in the context of IRE, targeted amplified the cDC1/CD8⁺ T cell axis. Here we also analyzed the effect of the hydrogel vaccine on tumor microenvironment without IRE therapy (**Figure for Reviewer 2 Comment 7**). The results showed that the hydrogel vaccine had minimal effect on pancreatic tumor microenvironment without IRE. We think that this misunderstanding may be caused by the ununiformed figure legends. In the revised manuscript, the figure legends labeling control and vaccine group are uniformed according to **Comment 4-5, Reviewer 3**. Thanks for the valuable suggestion, and we hope these revisions could clarify the issue.

Figure for Reviewer 2 Comment 7. The proportion of tumor-infiltrating CD4⁺ cells, CD8⁺ cells, Ly6G⁺ cells, CD103⁺CD11b⁻ cells and NK1.1⁺ cells across different groups.

Comment 8. To support the immunotherapeutic function of hydrogel microsphere

vaccines, the specificity against tumor cell type is essential. The authors should challenge the mice with distant tumors, which are different from primary tumor cell types.

Response: We thank the reviewer for the insightful suggestion. This suggestion has important implications for assessing whether the abscopal effects of the combination therapy are due to CD8⁺ T cell induced adaptive antitumor immunity for pancreatic cancers. On the basis of an orthotopic pancreatic cancer model, we induced distant metastasis using KL (murine squamous carcinoma) and Hepa1-6 (murine HCC cell line) cell lines. The results showed that the distant metastases formed by pancreatic cancer cell lines (KPC and Panc02) were significantly attenuated after the IRE and hydrogel vaccine combination therapy, whereas the distant metastases caused by KL and Hepa1-6 cells were not significantly shrunk. These results, together with the results of **Comment 12**, supported the highly adaptive anti-tumor immunity induced by the hydrogel microsphere vaccine. In the revised manuscript, these results were presented **Figure 7d-f, Supplementary Fig. S22c-d**. We hope this response will clarify this issue and meet your criteria.

Figure 7. Activated CD8⁺ T cells play a key role in the combination therapy-induced abscopal effect (d-f). (d) Representative images of H&E staining of lung metastasis of Panc02, KPC, KL and Hepa1-6 cells across different treatment groups; **(e)** The hydrogel vaccine combined with IRE ablation significantly the number of tumor nodes and tumor area in KPC and Panc02 induced lung metastases; **(f)** Representative images of IHC staining showed the expression of P63 in KL cells and the expression of AFP in Hepa1-6 cells.

Supplementary Fig.22. Representative images show the gross distant metastasis tumors across different groups (c-d). (c) Representative images of H&E staining of liver metastasis of Panc02, KPC, and Hepa1-6 cells across different treatment groups; **(d)** The hydrogel vaccine combined with IRE ablation significantly the tumour area in KPC and Panc02 induced liver metastases.

Comment 9. The authors need to provide in vitro and in vivo evidence of gene silencing of PD-L1 protein by Cas9-PD-L1 encoding plasmid DNA.

Response: Thanks for the valuable comment. Cas9-based genomic editing has been widely used by we [1, 2] and others, and the silencing of protein expression is the key point of gene knockout. In **Figure 20, p**, the in vivo results have provided clues that the

small Cas9 library targeting murine CD274 gene reduced the expression of PD-L1 in combination with IRE. To further validate the silence of PD-L1 protein, we performed in vitro IFN- γ induction experiments. IFN- γ is a well-recognized inducer of PD-L1 expression in tumor cells. The results showed that IFN- γ treatment for 24 and 48 hours could induce PD-L1 expression in Panc02 cells. After electroporation transfection of the Cas9 system, the expression of PD-L1 was reduced to extremely low levels (**Supplementary Fig. 23b**). This result demonstrates the efficient knockout of PD-L1 by the Cas9 system. We thanks again for the valuable comment.

Supplementary Fig. 23b. Western blot assays demonstrated the expression level of PD-L1 with IFN- γ treatment Panc02 cells.

Reference

- [1] Li Shuo, Yuan Bo, Cao Jixin et al. Docking sites inside Cas9 for adenine base editing diversification and RNA off-target elimination. [J]. Nat Commun, 2020, 11: 5827.
- [2] Cai Yuan, Cheng Tianlin, Yao Yichuan et al. In vivo genome editing rescues photoreceptor degeneration via a Cas9/RecA-mediated homology-directed repair pathway. [J]. Sci Adv, 2019, 5: eaav3335.

Comment 10. The authors should provide the intracellular expression of the Cas9 protein in the tumor cells and in vivo tumor tissues.

Response : Thanks for the valuable suggestion. In the revision, we performed immunofluorescence staining, WB assay and IHC staining to validate the expression of SpCas9 protein both in vitro and in vivo.

The direct immunofluorescence staining showed strong expression of SpCas9 protein in pancreatic cancers cells treated with electroporation transfection (**Figure for Reviewer 2 Comment 10a, right panel**). The Western Blot assay also showed the expression of SpCas9 in the electroporation transfected cells (**Figure for Reviewer 2 Comment 10b**). These results provide evidence to the expression of SpCas9 protein in vitro.

We have presented the expression of SpCas9 protein in vivo using Western Blot assay (**Figure 3p**). According to the reviewer's suggestions, to further verify the expression of SpCas9 in vivo, tissue staining was performed (**Figure for Reviewer 2 Comment 10c**). Since the existing SpCas9 antibody could not be used for staining the paraffin-embedded tissue, we validated the protein expression of SpCas9 by staining FLAG tag (**Figure for Reviewer 2 Comment 10c**). Compared with the non-IRE treated controls, the IRE treated tumor tissue showed significant expression of the FLAG protein tag surrounding the necrosis region. These results suggest that the SpCas9 protein is expressed in tumor tissues treated with the Cas9 plasmid and IRE ablation.

Figure for Reviewer 2 Comment 10. Validation of the expression of SpCas9 protein in vitro and in vivo. (a) The IF staining and **(b)** Western Blot assay showed strong expression of SpCas9 protein in pancreatic cancers cells treated with electroporation transfection; **(a)** IHC staining for FLAG label demonstrated the expression of SpCas9 in tumor tissue treated with Cas9 plasmid and IRE therapy.

Comment 11. The authors focused on the recruitment and amplification of cDC1s by FLT3L. How did the authors differentiate the recruited cDC1s from amplified cDC1s?

Response: Thanks for the valuable comment. To address this issue, we performed a series of flow cytometry and animal experiments. First, we analyzed the Ki67 expression of tumor-infiltrating cDC1s by flow cytometry. The proportion of Ki67⁺ cDC1s increased significantly after hydrogel microsphere vaccine therapy, indicating that the vaccine promoted amplification of cDC1s (**Figure for Reviewer 2 Comment 11. a-c**). Next, inspired by Gong et al. [1], we tested whether the hydrogel microsphere vaccine promoted recruitment of DC precursors by tail-vein injection of GFP-labeled BMDCs. In general, we isolated the bone marrow of Ai47 mice, and then transferred Cre-carrying plasmid to activate the expression of GFP. These BMDCs were then injected into mice bearing orthotopic pancreatic tumors via tail-vein injection. Flow cytometry analysis showed that local administration of hydrogel microsphere vaccine significantly increased the proportion of tumor-infiltrating GFP-labeled cells, suggesting that the vaccine increased the recruitment of circulating DC precursors into the tumor region (**Figure for Reviewer 2 Comment 11. a-c**). To some extent, these results demonstrated that the hydrogel microsphere vaccine promoted both local cDC1 amplification and DC recruitment.

However, due to individual variability, the diverse roles of cytokines, myeloid cell plasticity, and the complex nature of the immune microenvironment, the above-described methods are not completely precise in quantifying the recruitment and

expansion of tumor resident DCs. However, if the recruitment and expansion of cDC1s are considered as a continuous process, such results may, at least partially, explain the cDC1-induced antitumor immune activation during the hydrogel vaccine therapy. Finally, the precise depiction of this problem may still need the application of various genetic mouse models for lineage tracing and further research on immunological development. We have tried our best to clarify this issue, and we sincerely hope these results will meet your criteria.

Figure for Reviewer 2 Comment 11. (a) The represented images for analysis of GFP-labeled transfected DC precursors in pancreatic cancer in control group, IRE treated group and IRE combined with hydrogel vaccine group; (b) The represented images for analysis of tumor-infiltrating Ki67⁺cDC1s in pancreatic cancer in control group, IRE treated group and IRE combined with hydrogel vaccine group; (c) Barplots showed that IRE combined with hydrogel vaccine significantly increased the recruitment of GFP-labeled DC precursors as well as the Ki67⁺cDC1s in pancreatic cancer. Dunnett t-test was used to compare continuous variables between the experimental group and multiple control groups.

References

[1] Gong Zheng, Li Qing, Shi Jiayuan et al. Lung fibroblasts facilitate pre-metastatic niche formation by remodeling the local immune microenvironment. [J]. *Immunity*, 2022, 55: 1483-1500.e9.

Procedure for Generation of GFP-labeled BMDCs and tail vein transplantation experiment

The tail-vein injection of GFP labeled BMDCs were performed according to the protocol reported by Gong et al. with several modifications. The bone marrow cells were derived from the Ai47 mice. The procedure for harvesting the bone marrow cells has been described in detail in Methods section. The bone marrow cells were transfected with Cre-carrying plasmid (Addgene #49056) using high-voltage electroporation. About 5ug of the plasmid was used for transfection of 1×10^7 bone marrow cells diluted in 100ul $1 \times$ EBEP transfection buffer. The electroporation parameters are set as follows: 400 voltage, 800us duration, 2 pulse number. The transfection was considered successful with a GFP-positive cells over 50% percentage after 24 hours. The transfected cells were cultured using GM-CSF (20ng/ml) and IL-4 (10ng/ml) for three

days, and then low dose FLT3L and CD40L were added for co-incubation for 2 days. The death cells due to the electroporation can be reduced by changing the culture medium. The GFP-labeled BMDCs were injected in to mice bearing pancreatic cancer receiving different therapies. For mice receiving the combination therapy of IRE and hydrogel microsphere vaccine, 1×10^7 pre-induced BMDCs were immediately tail-vein injected after the combination therapy. Subsequently, the GFP-labeled cells were injected for 3 consecutive days. The tumors were harvested in the fifth day for flow cytometry analyses of GFP-positive cells.

Comment 12. The authors should test which immune cells in the tumor microenvironment are the major cell types for preventing the growth of distant tumors.

Response: We thank the reviewer for the insightful comment. Considering the amplification of the cDC1/CD8⁺ T cell cascade induced by the hydrogel microsphere vaccine, we first examined the infiltration of DC1 and CD8⁺ T cells in the metastases. Flow cytometry analysis showed that there was no significant difference in the number of infiltrating CD45⁺ cells across the IRE, IRE combined with α PD-L1, and IRE combined with hydrogel vaccine group (**Figure 7a**). Further results suggested that the proportion of CD103⁺CD11b⁻ cDC1s in subcutaneous tumors did not change significantly, indicating that cDC1 may not be a direct effector inducing the abscopal effects (**Figure 7b**). Considering that previous study reported that CD40L may also have a function on tumor-infiltrating macrophages [1], the F4/80⁺CD11b⁺ total macrophages,

CD206⁺CD80⁺ and CD163⁺CD80⁺ M2-like macrophages were then analyzed. The results showed that *in situ* administration of hydrogel vaccine had no significant effect on the proportion and phenotype of macrophages in the distant subcutaneous tumors (**Figure 7b**). Finally, the infiltration and function of CD8⁺ T cells were analyzed. After the hydrogel vaccination, there is a significant increase in the proportion of infiltrating IFN- γ ⁺CD8⁺ T cells in the subcutaneous tumors (**Figure 7c**). Since IFN- γ is one of the important cytokines for evaluating cytotoxic T cell function, these results may suggest that CD8⁺ T cells are a major player in the distant effect. Taken together, these results demonstrated that the activated CD8⁺ T cells are direct effector immune cells causing the abscopal effect. These results were present in **Figure 7a-c** and corresponding manuscript (**Page 13, line 27-29; Page 14**).

Figure 7. Activated CD8⁺ T cells play a key role in the combination therapy-induced abscopal effect (a-c). (a) Flow cytometry analysis revealed no differences in the count of tumor-infiltrating CD45⁺ cells; (b) Flow cytometry analysis revealed no differences in proportion of cDC1 cells, M1 and M2 cells in distant subcutaneous tumors across different treatment groups. (c) Flow cytometry analysis

showed that the proportion of IFN- γ ⁺CD8⁺ T cells in distant tumors increased significantly in subcutaneous pancreatic tumors;

References

[1] Jiang Honglin, Courau Tristan, Borison Joseph et al. Activating Immune Recognition in Pancreatic Ductal Adenocarcinoma via Autophagy Inhibition, MEK Blockade, and CD40 Agonism. [J]. *Gastroenterology*, 2022, 162: 590-603.e14.

Minor issues

Comment 13. In line 489, “The plasmid-to-lip ratio is 1:1, 1:1.5, 1:2, 1:2.5 and 1:3 according to the experimental design”. The authors should clarify the unit of plasmid DNA to lipid ratios. Is it the N/P ratio or weight ratio?

Response: We thank the reviewer for the valuable comment. The plasmid-to-lip ratios (1:1, 1:1.5, 1:2, 1:2.5 and 1:3) described in the manuscript are w/v ratios. According to the suggestion, we have clarified the unit of plasmid DNA to lipid ratios in the revised manuscript (**Methods, section “Detailed information for Preparation of the hydrogel microsphere vaccine”, Page 17, line 25-29**). The detailed amounts of plasmid and Liposome corresponding to each ratio in the pilot experiments are listed as follow:

The detailed amounts of plasma and Liposome corresponding to each ratio in the pilot test.

	1:1	1:1.5	1:2	1:2.5	1:3
pDNA	6 μ g	6 μ g	6 μ g	6 μ g	6 μ g
Liposome	6 μ L	9 μ L	12 μ L	15 μ L	18 μ L

Comment 14. The authors should describe which plasmid-to-liposome ratio was used for the final hydrogel system.

Response: Thanks for the kind reminder. We sincerely apologize for our carelessness. The plasmid-to-lip ratio (w/v) used in the final hydrogel system was 1:1 (please also refer to **Comment 13.**), and the concentration of Cas9 plasmid was 50 ng/ul in the final system. We have added the information in the revised manuscript (**Methods, section “Detailed information for Preparation of the hydrogel microsphere vaccine”, Page 17, line 25-29**).

Comment 15. In the method section, what is the lipid composition of the cationic liposome?

Response: Thanks for the valuable comment.

The DLin-MC3-DMA/Cholesterol/DSPC/PEG2000-DSPE system was composed at a molar ratio of 52/38.5/8/1.5. The detailed preparation process is described as follows and has been added to the Methods section in the revised manuscript (**Methods, Page 17, line 18-20**).

To prepare the Plasmid@Lip nanoparticles, a methanol solution of 10 mM DLin-MC3-DMA/Cholesterol/DSPC/PEG2000-DSPE at a molar ratio of 52/38.5/8/1.5 and 1×PBS solution containing Cas9 plasmid were mixed using a microfluidic device at a flow of 10 mL/min with a flow rate ratio of 3:1 (aqueous to lip phase). The mixed solution was left standing for 30 minutes at room temperature. The mixture was then transferred into a dialysis bag (interception molecular weight: 3500 Da) for 24 hours of

dialysis. The plasmid-to-lip ratio (w/v ratio) is 1:1, 1:1.5, 1:2, 1:2.5 and 1:3 according to the experimental design.

Reviewer 3

In this manuscript, Liu et al. develop and study the effects of a hydrogel microsphere vaccine in the treatment of pancreatic cancer. The authors report that treatment of orthotopic KPC pancreatic mouse models with the hydrogel microsphere vaccine increased animal survival and inhibited the growth of distant metastasis. This effect is due to an activation of cDC1 cells and consequent antigen presentation and activation of CD8⁺ T cells.

Overall, the manuscript is well written, well positioned in the state-of-the-art of the field and the authors provide clear assays that support their claims making this manuscript a good addition to the literature. However, the data presented does not fully support the conclusions and some issues need to be addressed to clarify the results.

Major issues

Comment 1. Regarding the data in figure 2p and 2q, the authors should check if the hydrogel microsphere vaccine had any effects on cDC1s' proliferation and differentiation.

Response: Thanks for the valuable suggestion. To clarify this issue, the proliferation and differentiation status of bone-marrow derived cells were analyzed by flow cytometry analyses. First, the results showed that hydrogel microsphere vaccine increased the proportion of Ki67-positive cells in cDC1s, suggesting that it had a promotion effect of the cDC1s proliferation (**Supplementary Fig. 4a**).

Considering the potential role of CD40L on macrophages, we also conducted a

preliminary analysis of the bone-marrow derived macrophages. Positive expression of CD206 or CD163 is used as independent marker to distinguish M2-like macrophages. The results showed that the additional CD40L or FLT3L slightly promoted the differentiation of BMDC into M1-like macrophages and cDC1s (**Supplementary Fig. 4b, c**). The sequential administration of FLT3L and CD40L, including the hydrogel microsphere vaccine, promoted the differentiation of BMDC into cDC1 to a greater extent and kept the higher expression level of Ki67 in these cDC1s (**Supplementary Fig. 4a**). These results were also supplemented in the revised manuscript (**Page 7, line 13-17**).

Supplementary Fig. 4. The proliferation and differentiation of bone marrow derived cells towards cDC1s and macrophages (a-c). (a) Analysis of the effect of hydrogel microsphere vaccines and the counterparts on

the differentiation and proliferation in cDC1s; **(b)** Analysis of the effect of hydrogel microsphere vaccines and the counterparts on the differentiation of BMDC into macrophages. The CD163⁺CD80⁺ macrophages and CD206⁺CD80⁺ macrophages represented the M2-like macrophages, and the CD163⁻CD80⁺ macrophages and CD163⁻CD80⁻ macrophages represented the M1-like macrophages.

Comment 2. On figure 3k and 4b the authors show IHCs for CD45 on mouse pancreatic tumors to assess whether IRES ablation and the hydrogel microsphere vaccine affected immune cell infiltration in orthotopic pancreatic tumors. Because the study focuses on cDC1s and CD8⁺ T cells, the authors should also perform staining for these immune cell types. This would allow the assessment of whether IRES ablation and the hydrogel microsphere vaccine being tested are able to promote the recruitment of these specific immune cells into the TME.

Response: We thank the reviewer for the valuable suggestion. According to the suggestion, we performed the immunofluorescence staining assays and the results were presented in **Supplementary Fig.9 b, c.**

Supplementary Fig. 9. Analysis the tumor necrosis, CD8⁺ and CD103⁺ immune cell infiltration across

different treatment groups (b, c). (b-c) Representative image of immunofluorescence staining represented the CD8⁺ and CD103⁺ immune cell infiltration across different treatment groups.

Due to the limitations of antibodies and fluorescence channels, CD103⁺ cells were identified as potentials cDC1s, and CD8⁺ cells were considered as CD8⁺ T cells. The results of immunofluorescence staining were highly consistent with those of flow cytometry analyses in **Figure 4**. The IRE ablation combined with the hydrogel microsphere vaccine substantially promoted the immune cell infiltration into orthotopic pancreatic tumors (**Figure 4, Supplementary Fig. 9b, c**). In addition, the results of the immunofluorescence staining also showed that a large number of CD103⁺ and CD8⁺ cells were recruited into the peripheral and marginal regions of the tumor after the combination therapy (**Supplementary Fig. 9b, c**). Since flow cytometry is more accurate for cell qualitative and quantitative analysis, we still use the quantitative analysis results of flow cytometry in Figure 4. We hope these additional experiments will meet your criteria.

Comment 3. On line 289, while describing figure 4c, I believe the authors mistakenly wrote FLT3L, as in the figure it is the CD40L's highest concentration that lasted from 48 hours to 96 hours.

Response: We thank the reviewer for the kind reminder. We sincerely apologize for our carelessness. We have checked the figure panels and the corresponding text and corrected the error (*Page 10, line 3-4*).

Comment 4. In Figure 4b, the hydrogel microsphere vaccine group is labelled as “IRE+Hydrogel microspheres vaccine”, while in figures 4e, 4f, 4k and 4l it is labelled as “Group 5”. Please change these labelled to something more uniform. Also, I believe calling it Group 5 might cause confusion as all the other groups are controls.

Response: Thanks for the kind reminder. We sincerely apologize for our carelessness. We think this issue is similar to **Comment 5**, where confusion arises due to the inconsistent labeling of control and treatment names. In the revised manuscript, we have uniformed the name of the hydrogel microsphere vaccine and different controls. In the revised manuscript, the hydrogel microsphere vaccine group is labeled as “hydrogel microsphere vaccine” or “hydrogel vaccine” in relevant figures. We hope that this revision will minimize the possibility of confusion. All of these changes can be traced in the manuscript with the change track file.

Comment 5. Please explain, in the results section, the treatment groups presented on figures 4i, 4j, 4m and figures 5a-5e. Are these the same as Groups 1 to 4 presented in other figures? This should be clarified and the names given should be simplified in a way that the reader immediately knows to which group the authors are referring while describing the results.

Response: Thanks for the kind reminder. We sincerely apologize for our carelessness. The treatment groups presented on figures 4i, 4j, 4m and figures 5a-5e are the same as Groups 1 to 4 presented in other Figure 5. We think this issue is similar to **Comment**

4, where confusion arises due to the inconsistent labeling of control and treatment names. In the revised manuscript, we have uniformed the name of the hydrogel microsphere vaccine and different controls. The hydrogel microsphere vaccine group is labeled as “hydrogel microsphere vaccine” or “hydrogel vaccine” in relevant figures and manuscripts. The names of the hydrogel counterparts and their compositions were stressed in the **Methods (Page 18, line 7-10)**, which also listed as follow:

Group1 represented the **CaCO₃/Cas9@Lip/@HAMA** counterpart.

Group2 represented the **Flt3L/Cas9@Lip/@HAMA** counterpart.

Group3 represented the **CD40L/Cas9@Lip/@HAMA** counterpart.

Group4 represented the **CD40L/Flt3L/Cas9@Lip/@HAMA** counterpart.

In the revision, these names are uniformly used in the barplot across **Figure 4-5**. The composition names are presented in representative images of staining and flow cytometry experiment. We hope that this revision will minimize the possibility of confusion. All of these changes can be traced in the manuscript with the change track file. Thanks again for the kind reminder.

Comment 6. In Materials and Methods, the authors say that tumor volume was evaluated every 3 days, but the method used is not mentioned. Also, in the experiment depicted in figure 6, did the authors assess tumor volume and weight at the time of euthanasia? Was there any difference in tumor volume between groups?

Response: Thanks for the kind comment. To evaluate the subcutaneous tumor volumes,

the maximum and minimum axis of the subcutaneous tumors were evaluated using a caliper after the mice were isoflurane-anesthetized. The tumour volume (V) was calculated using the following formula: tumour volume = $0.52 \times L \times W^2$ (L=the major tumour axis; W= the minor tumour axis). The approach was also reported in our previous study [1, 2]. In **Figure 6**, the tumor volumes were assessed after isoflurane anesthesia. Meanwhile, the tumor weights were assessed after the scarification of the mice and split of the tumors at the end point. We have added the detailed information in the **Methods** section (**Page 20, line 20-21**). **Figure 6k** demonstrated the differences in tumor volume between groups at the end point. The results showed that the tumour volumes of the IRE combined with hydrogel microsphere vaccine group were significantly smaller than that of the other groups. We hope these revisions will be to your satisfaction.

References

- [1] Liu Xiaoyu, Li Zhi, Wang Zhongmin et al. Chromatin Remodeling Induced by ARID1A Loss in Lung Cancer Promotes Glycolysis and Confers JQ1 Vulnerability. [J]. *Cancer Res*, 2022, 82: 791-804.
- [2] Ren Xinxin, Rong Zhuoxian, Liu Xiaoyu et al. The Protein Kinase Activity of NME7 Activates Wnt/ β -Catenin Signaling to Promote One-Carbon Metabolism in Hepatocellular Carcinoma. [J]. *Cancer Res*, 2022, 82: 60-74.

Comment 7. In Materials and Methods, it is not mentioned how the metastasis was evaluated. Also, the authors mention on line 406 that the metastases that are being evaluated are “distant subcutaneous metastases.” Orthotopic pancreatic

mouse models don't develop subcutaneous metastasis but rather liver and lung metastasis, and sometimes peritoneal. dissemination can also be seen. Could you please clarify this issue?

Response: We thank the reviewer for the insightful comment. We understand the reviewer's concerns and believe that this issue is important and relevant to the clinical translation of the novel therapeutic strategy.

First, the evaluation of the subcutaneous metastases was according to the tumor volume and weight. The assessment of the tumor volume has been described in **Comment 6** and added in the **Method** section.

Next, to further clarify the issue, we introduced a set of pancreatic cancer lung metastases and liver metastases models. Briefly, the lung metastases were established by tail-vein injection of tumor cells, while the liver metastases were established by semi-spleen injection of tumor cells [1]. Panc02 and KPC pancreatic cells were used to mimic the lung and liver metastasis of pancreatic cancers. To further test whether the immunotherapeutic effect of the combination therapy is specific for pancreatic cancers, KL (murine lung squamous cell carcinoma) and Hepa1-6 (murine HCCs) cell lines were also used to establish the lung or liver metastasis models (**Reviewer 2, Comment 8**). We compared the abscopal effect across these metastasis models.

Consistent with the results of subcutaneous tumors shown in **Figure 6h-l**, the combination therapy also effectively attenuated the formation and progression of lung or liver metastases of KPC and Panc02 cells, while no significant attenuation of the distant metastasis of KL and Hepa1-6 tumors was observed (**Figure 7a-e**,

Supplementary Fig. 18 c-d). It should be noted that the efficiency of the abscopal effect may differ in lung and liver metastatic models. We proposed that the combination therapy enhanced the abscopal effect by promoting an increase in the proportion of activated CD8⁺ T cells rather than by enhancing the infiltration of cDC1s, or M1-like macrophages in distant tumors (**Reviewer 2, Comment 12**) [2,3]. The various immunological milieu of the target organs may also involve the formation of abscopal effect.

In conclusion, these results strengthened the conclusion that IRE ablation combined with hydrogel microsphere vaccine could induce a significant abscopal effect in a preclinical pancreatic cancer model and also suggested that the abscopal effect was amplified by adaptive antitumor immunity specifically against pancreatic cancers. We hope you will find these revisions satisfactory.

Figure 7. Activated CD8⁺ T cells play a key role in the combination therapy-induced abscopal effect (d-f). (d) Representative images of H&E staining of lung metastasis of Panc02, KPC, KL and Hepa1-6 cells across different treatment groups; **(e)** The hydrogel vaccine combined with IRE ablation significantly the number of tumor nodes and tumor area in KPC and Panc02 induced lung metastases;

(f) Representative images of IHC staining showed the expression of P63 in KL cells and the expression of AFP in Hepa1-6 cells.

Supplementary Fig.18. Representative images show the gross distant metastasis tumors across different groups (c-d). (c) Representative images of H&E staining of liver metastasis of Panc02, KPC, and Hepa1-6 cells across different treatment groups; (d) The hydrogel vaccine combined with IRE ablation significantly the tumor tumour area in KPC and Panc02 induced liver metastases.

References

- [1] Soares Kevin C, Foley Kelly, Olino Kelly et al. A preclinical murine model of hepatic metastases. [J]. J Vis Exp, 2014, undefined: 51677.
- [2] Wang Weimin, Green Michael, Choi Jae Eun et al. CD8⁺ T cells regulate tumour ferroptosis during cancer immunotherapy. [J]. Nature, 2019, 569: 270-274.
- [3] Liao Peng, Wang Weimin, Wang Weichao et al. CD8 T cells and fatty acids orchestrate tumor ferroptosis and immunity via ACSL4. [J]. Cancer Cell, 2022, 40: 365-378.e6.

Comment 8. The authors never mention some important immune cells that are crucial in the TME and inflammation response, like macrophages. Analysis for these innate immune cells and, additionally, if these cells change fate - from pro to antitumorigenic, or vice-versa, especially in the survival studies, should be included.

Response: Thanks for the valuable suggestion. We understand the concerns of the reviewer. The general status of Ly6G⁺ macrophages and NK cells has been described in **Figure 5e**. According to the suggestion of the reviewer, we introduced new panels of flow cytometry experiments to assess the phenotype of tumor-infiltrating macrophages across different treatment groups.

The results showed that the hydrogel microsphere vaccine slightly increased the abundance of total macrophages, which mainly caused a slight increase in the proportion of M1-like proinflammatory macrophages. This may be related to the activation effect of the CD40 agonist on M1-like macrophages [1]. We also observed a trend of M1-like macrophage activation in the groups with CD40 agonist, although it was not statistically

significant across some groups (Supplementary Fig. 17a). However, according to current reports, M1-like macrophages commonly do not induce immunosuppressive effects to attenuate the effect of immune therapy. The flow cytometry gating strategy for the phenotype of macrophages were also provided in Supplementary Fig. 17b. The detailed information on the antibodies for macrophage analysis was also provided in the **Methods (Page 25, line 6-12)**. We hope you will find these supplementary experiments and explanations satisfactory.

Supplementary Fig. 17. Flow cytometry gating strategy for immune phenotyping of the macrophages. (a) Statistical analysis revealed no differences in the proportion of total F4/80⁺ macrophages in orthotopic pancreatic tumours across different treatment groups, while the M2-like immunosuppressive macrophages (CD163⁺CD80⁺ macrophages and CD206⁺CD80⁺ macrophages) slightly reduced after the CD40L and FLT3L

combination treatment. **(b)** The flow cytometry gating strategy for macrophages. The CD206⁺CD80⁺ and CD163⁺CD80⁺ macrophage represented for the M2-like macrophages, and the CD206⁻CD80⁺ and CD163⁻CD80⁺ macrophage represented for the M2-like macrophages

Reference

[1] Jiang Honglin, Courau Tristan, Borison Joseph et al. Activating Immune Recognition in Pancreatic Ductal Adenocarcinoma via Autophagy Inhibition, MEK Blockade, and CD40 Agonism. [J]. *Gastroenterology*, 2022, 162: 590-603.e14.

Minor issues

Comment 9. Figure S7 misses legends for panels c and d.

Response: Thanks for the kind reminder. We sincerely apologize for our carelessness.

The figure legends have been added in the revised Figure S7.

Comment 10. Figure 5k can be deleted. I believe it is a repetition of 1d.

Response: Thanks for the kind reminder. Figure 5k has been deleted in the revised manuscript.

REVIEWER COMMENTS

Reviewer #1 (Remarks to the Author):

I am satisfied with the responses to my original comments.

Reviewer #3 (Remarks to the Author):

Thank you for your comprehensive responses to the reviewers' comments. The authors have made all the necessary changes in the manuscript to address the issues pointed out by the reviewers and to bring more depth into the study. An effort was made to improve the manuscript with all the information requested and it greatly improved the manuscript regarding the use of IRE therapy in combination with hydrogel microsphere vaccine. The improved Figures and Tables provided also further complement the manuscript and help the reader fully understand the information included in the text.

Reviewer #4 (Remarks to the Author):

Please see below my feedback on the authors' response to the comments of Review 2.

1. Comment 1. Regarding the innovation of this work, the authors responded with two points. Firstly, they claim that the targeting of cDC1 for priming CD8+ with synthetic materials is innovative "However, there is still no effective strategy to locally activate and augment cDC1-mediated anti-tumor immunity". After a small literature search, it became clear that they have already been reports on synthetic materials for the activation of cDC1 to provoke immunity, please see below a few examples

(<https://www.ncbi.nlm.nih.gov/pmc/articles/PMC9414227/pdf/pharmaceutics-14-01614.pdf>,
<https://pubs.acs.org/doi/pdf/10.1021/acs.molpharmaceut.0c00984>,
[https://www.cell.com/molecular-therapy-family/methods/fulltext/S2329-0501\(19\)30123-8#](https://www.cell.com/molecular-therapy-family/methods/fulltext/S2329-0501(19)30123-8#) ,
<https://www.mdpi.com/2076-393X/9/1/56>).

The second innovative feature of the work is the application of synthetic materials for the treatment of pancreatic tumors. However, biomaterials approaches for treating pancreatic tumors, either via immunotherapy or chemotherapy have been quite frequently reported, a few examples are listed here: <https://pubs.acs.org/doi/full/10.1021/acs.nanolett.2c01994>,
<https://pubs.rsc.org/en/content/articlehtml/2016/ra/c6ra07934b>,
<https://link.springer.com/article/10.1007/s12274-019-2342-7>,
<https://pubs.acs.org/doi/full/10.1021/acs.biomac.8b00959>.

2. Comment 2. Regarding the controlled release of CD40L from the hydrogel microspheres, Supplementary Figure 2a shows that the release rate between pH 6.8 and 7.4 is very minor, considering the relatively high SD values of the data points, the difference seems to be insignificant, which is also not tested by the authors. This small difference can be barely considered useful as a mechanism to trigger tumor-specific payload release.

3. Comment 3. The authors claimed that Ca ions were loaded into the micelles via complexation with carboxylic groups. This raised a fundamental question of the micelle formation: with the hydrophilic carboxylic groups at the end of the hydrophobic chain end of PLGA-PEG-PLGA, this will affect micelles formation as the micelles are self-assembled via hydrophobic interactions. This question is neglected but is fundamental in the design of the materials.

4. Comment 5. The authors claimed "The specific preparation process was to first load CD40L into HOOC-PLGAPEG-PLGA-COOH micelles, and then form calcium carbonate nanoparticles around the micelles through biomineralization." However, this is another fundamental question that needs to be validated. Because it is known that hydrophilic biomolecules such as CD40L do not load in micelles which normally encapsulate payloads via hydrophobic interactions.

5. Comment 6. The pH difference before and after the IRE therapy was really small (~6.8 vs. ~6.7) to be used as a trigger for tumor-specific drug release, as mentioned in comment 2. Besides, even the analysis method error can be higher than this small pH change. Therefore, the authors should also show that their method is accurate enough to distinguish the small pH change before and after IRE treatment.

6. Comment 12. Regarding the question "which immune cells in the tumor microenvironment are the major cell types for preventing the growth of distant tumors", The authors addressed this question by quantifying immune cell populations in tumors. This is not a reliable method and the data are not convincing. Instead, the authors should perform knock-out/knock-in experiments in which they deplete the target cells and then adoptively infuse the cells to confirm whether the cDC1 induced the anti-tumor immunity. What was also questionable is that cDC1, as antigen-presenting cells, should play their role in secondary lymphoid organs to activate CD8 T cells. This is another reason why the tumor immune cell counting in this case is not able to fully address this question.

[REDACTED]

Point-to-point response to reviewers

Reviewer #4

1-Comment 1. Regarding the innovation of this work, the authors responded with two points. Firstly, they claim that the targeting of cDC1 for priming CD8⁺ with synthetic materials is innovative “However, there is still no effective strategy to locally activate and augment cDC1-mediated anti-tumor immunity”. After a small literature search, it became clear that they have already been reports on synthetic materials for the activation of cDC1 to provoke immunity, please see below a few examples

<https://www.ncbi.nlm.nih.gov/pmc/articles/PMC9414227/pdf/pharmaceutics-14-01614.pdf>,

<https://pubs.acs.org/doi/pdf/10.1021/acs.molpharmaceut.0c00984>,

[https://www.cell.com/molecular-therapy-family/methods/fulltext/S2329-0501\(19\)30123-8#](https://www.cell.com/molecular-therapy-family/methods/fulltext/S2329-0501(19)30123-8#) ,

<https://www.mdpi.com/2076-393X/9/1/56>). The second innovative feature of the work is the application

of synthetic materials for the treatment of pancreatic tumors. However, biomaterials approaches for treating pancreatic tumors, either via immunotherapy or chemotherapy have been quite frequently reported, a few examples are listed here: <https://pubs.acs.org/doi/full/10.1021/acs.nanolett.2c01994>,

<https://pubs.rsc.org/en/content/articlehtml/2016/ra/c6ra07934b>,

<https://link.springer.com/article/10.1007/s12274-019-2342-7>,

<https://pubs.acs.org/doi/full/10.1021/acs.biomac.8b00959>.

Response: We are grateful to the reviewer for the constructive comments. We fully understand the concerns of the reviewer. As the reviewer noted, there have been previous reports of using biomaterials to treat pancreatic cancer, as well as original studies employing biomaterials to activate cDC1s. It also demonstrates that, as a deadly digestive system tumour, pancreatic cancer has always been the focus of clinical and basic research. In addition to immune checkpoint blockade, the activation of antigen-presenting cells has been a frontier of immunotherapy for the past decade. However, due to our limited understanding of the microenvironment of pancreatic cancer and the regulatory mechanism of antigen-presenting cells in tumour microenvironment, it is challenging to effectively activate antigen-presenting cells to fight against pancreatic tumours.

As we have explained previously, it is because of recent advances in oncology and immunology that we could attempt to efficiently activate the antitumor immunity of cDC1s in pancreatic cancer through new combination therapy and ultimately achieve a significant therapeutic effect *in vivo*. Considering the current clinical predicament in the treatment of advanced pancreatic cancers, we believe this new strategy gains novelty and has potential for clinical translation from the perspective of oncology. We thank the reviewer

again for the valuable comment. We sincerely hope this explanation can clarify this issue.

2-Comment 2. Regarding the controlled release of CD40L from the hydrogel microspheres, Supplementary Figure 2a shows that the release rate between pH 6.8 and 7.4 is very minor, considering the relatively high SD values of the data points, the difference seems to be insignificant, which is also not tested by the authors. This small difference can be barely considered useful as a mechanism to trigger tumor-specific payload release.

Response: We thank the reviewer for the valuable comments. We apologize for the confusion caused by the imperfect analysis of the experimental data. We have added statistical comparisons of differences in CD40L release on the same day at different pH values to the newly provided **Supplementary Fig. 2a**. The raw data is also provided in the **Source Data** file. The results showed that pH level had minimal effect on drug release on day 0-2 and that there was no significant difference between CD40L release levels at different pH levels. However, there was a statistically significant difference in drug release at different pH levels over the next 6 days. The release of CD40L from CaCO₃ nanoparticles was designed for two primary purposes in our study. To complete the biological function of sequential regulation of cytokines, it was necessary to ensure that the early release of CD40L was slower than that of FLT3L (relevant results can be found in **Fig. 2i**). Next, as shown in **Supplemental Figure 2a**, the amount of CD40L released on day 6 at a pH of 7.4 is negligible, which is a waste of valuable drugs. The current results suggest that the use of CaCO₃ nanospheres is adequate to fulfil the necessary requirements of CD40L loading and release.

Supplementary Figure 2a. The release of CD40L from CD40L@CaCO₃/Flt3L/Cas9@lip/HAMA hydrogel microspheres at pH value of 6.8 and 7.4 in vitro.

3-Comment 3. The authors claimed that Ca ions were loaded into the micelles via complexation with carboxylic groups. This raised a fundamental question of the micelle formation: with the hydrophilic carboxylic groups at the end of the hydrophobic chain end of PLGA-PEG-PLGA, this will affect micelles formation as the micelles are self-assembled via hydrophobic interactions. This question is neglected but is fundamental in the design of the materials.

Response: We thank the reviewer for the very insightful comment. We understand the concerns of the reviewer. The terminal micelles are one of the critical factors influencing the formation of stable micellar structures in amphiphilic block polymers such as PLGA-PEG-PLGA. The mechanism by which amphiphilic polymers form micelles is that when the polymer reaches a certain concentration in the water system, the molecules self-assemble to form an ordered arrangement of thermodynamically stable structures. Due to the presence of hydrophobic groups, the force of repulsion between water molecules and polymers is greater than the force of attraction. Under the influence of van der Waals force, the hydrophobic portion forms the micellar core, while the hydrophilic group forms the micellar's outer layer, which is stably dispersed in aqueous solution. Although the modification of the terminal carboxylation enhances the hydrophilicity of PLGA-PEG-PLGA, the combination of PLGA with a hydrophobic segment is relatively hydrophobic overall. In a previous study [1], Prof. Ding and Yu's research group conducted a comprehensive analysis of the self-assembly behaviour of PLGA-PEG-PLGA containing terminal carboxyl groups. In **Figure 10** of the report, the authors demonstrated the ability of PLGA-PEG-PLGA modified with a carboxyl group to self-assemble into micelles at various pH values. In our manuscript, the results of **Figure 2 (h)** also presented that stable nano-micelles could still be formed after carboxyl modification. The carboxyl groups in the micelle core can complex with supersaturated Ca^{2+} in solution to provide nucleation sites, while Ca^{2+} concentrates around the nucleation sites to increase the local supersaturation, while spontaneously forming aggregates. Under the action of Ca^{2+} bridging and hydrogen bonding, the aggregates rearrange and self-assemble to form calcium carbonate nanoparticles [2, 3]. In the process of carboxyl groups forming nanoparticles with Ca^{2+} complex, micelles only act as a template for polycarboxyl groups at the beginning to promote Ca^{2+} complexation and aggregation. As the reviewer mentioned, this problem is a fundamental problem in material synthesis. However, we regret not having discovered a kind a direct method for determining whether the micelle still exists after complexation. We strongly agree with the reviewer's opinion. At the same time, we also believe that based on existing theories and evidence, it is reasonable to explain the synthesis mechanism of our material.

References:

- [1] Chang G., Yu L., Yang Z., et al. A delicate ionizable-group effect on self-assembly and thermogelling of amphiphilic block copolymers in water[J]. *Polymer*, 2009, 50(25):6111-6120.
- [2] Gao Y., Yu S., Cong H., et al. Block-copolymer-controlled growth of CaCO₃ microrings. [J]. *J Phys Chem B*, 2006, 110: 6432-6436.
- [3] Chen W., Wang G., Yung B., et al. Long-Acting Release Formulation of Exendin-4 Based on Biomimetic Mineralization for Type 2 Diabetes Therapy. [J]. *ACS Nano*, 2017, 11: 5062-5069.

4-Comment 5. The authors claimed “The specific preparation process was to first load CD40L into HOOC-PLGA-PEG-PLGA-COOH micelles, and then form calcium carbonate nanoparticles around the micelles through biomineralization.” However, this is another fundamental question that needs to be validated. Because it is known that hydrophilic biomolecules such as CD40L do not load in micelles which normally encapsulate payloads via hydrophobic interactions.

Response: We thank the reviewer for the valuable comment. We apologize for the confusion caused to the reviewers as a result of our inadequate explanation of the loading mechanism of CD40L in the paper. CD40L is loaded into CD40L@CaCO₃ nanoparticles via two mechanisms. First, CD40L cytokine is a kind of protein with hydrophilic and hydrophobic regions in its polypeptide sequence. The first mechanism is hydrophobic interaction between the hydrophobic region of CD40L protein and the hydrophobic chain segment of PLGA-PEG-PLGA. Another mechanism is that CD40L can be mineralized to form nanosized mineral solids by means of the reaction between acidic amino acid residues and calcium ions in a supersaturated environment with negligible influence on peptide bioactivity. These are the two primary mechanisms by which CD40L is loaded into CaCO₃ nanoparticles. The use of CaCO₃ nanoparticles loaded with small molecular proteins to achieve gradual dissolution and release of encapsulated proteins in the acidic and inflammatory microenvironment of tumours has been validated and implemented in several other previous studies [1-2]. Thanks for the valuable suggestion, and we hope these revisions could clarify the issue.

References:

- [1] Chen Q., Wang C., Zhang X., et al. In situ sprayed bioresponsive immunotherapeutic gel for post-surgical cancer treatment. [J]. *Nat Nanotechnol*, 2019, 14: 89-97.
- [2] Yang Z., Zhu Y., Dong Z., et al. Tumor-killing nanoreactors fueled by tumor debris can enhance radiofrequency ablation therapy and boost antitumor immune responses. [J]. *Nat Commun*, 2021, 12: 4299.

5-Comment 6. The pH difference before and after the IRE therapy was really small (~6.8 vs. ~6.7) to be used as a trigger for tumor-specific drug release, as mentioned in comment 2. Besides, even the analysis method error can be higher than this small pH change. Therefore, the authors should also show that their method is accurate enough to distinguish the small pH change before and after IRE treatment.

Response: Thanks for the valuable suggestion. In the original manuscript, we have explained that the tumour acidic microenvironment is a driving force that triggers the release of CaCO₃-loaded drugs. In this study, the acidic microenvironment of pancreatic cancer itself rather than the difference in pH before and after treatment drives the drug release.

In the initial round of peer review, Reviewer 2 believed that there was no evidence indicating the pH change before and after tumour ablation therapy, so he suggested that we confirm that the tumour microenvironment remained acidic after ablation in order to facilitate the sustained drug release. According to the suggestion, we measured the pH value of the tumour microenvironment and found that pH did not elevate after the ablation therapy. This result was consistent with previous reports [1-2]. Thus, the current measurement accuracy was sufficient to indicate that the tumour microenvironment after ablation therapy could still drive the release of drugs in CaCO₃. We believe that this problem is mainly due to the interpretation of the manuscript text, rather than a technical problem.

We appreciate the reviewer's suggestion which enables us to further clarify this issue.

References:

- [1] Nikfarjam Mehrdad, Muralidharan Vijayaragavan, Christophi Christopher, Mechanisms of focal heat destruction of liver tumors. [J]. J Surg Res, 2005, 127: 208-23.
- [2] Song C W, Kang M S, Rhee J G et al. The effect of hyperthermia on vascular function, pH, and cell survival. [J]. Radiology, 1980, 137: 795-803.

6-Comment 12. Regarding the question “which immune cells in the tumor microenvironment are the major cell types for preventing the growth of distant tumors”, The authors addressed this question by quantifying immune cell populations in tumors. This is not a reliable method and the data are not convincing. Instead, the authors should perform knock-out/knock-in experiments in which they deplete the target cells and then adoptively infuse the cells to confirm whether the cDC1 induced the anti-tumor immunity. What was also questionable is that cDC1, as antigen-presenting cells, should play their role in secondary lymphoid organs to activate CD8 T cells. This is another reason why the tumor immune cell

counting in this case is not able to fully address this question.

Response: We thank the reviewer for the insightful comment. In the original version of the manuscript, we have recognized the importance of secondary lymphoid organs, such as tumour-draining lymph nodes (TdLNs), in the activation and amplification of antitumor immunity. Thus, we first analyzed the activation and migration of cDC1s in TdLNs after different treatment combinations, and found that the migration of cDC1s into TdLNs was significantly enhanced following a combination of IRE and hydrogel vaccine. In addition, the enrichment of cDC1s in TdLNs was closely related to the activation of CD8⁺ T cells after the combination therapy.

In addition to preventing local tumour progression, we observed that the combination therapy also induced an abscopal effect. According to the comments of reviewer 2, we evaluated the potential immunological mechanism that may be responsible for the abscopal effect. It has been reported that the activation of various immune cells, such as CD8⁺ T cells, antitumour macrophages, and NK cells, can lead to abscopal effect in tumour therapies [1-4]. Considering the local enhancement of cDC1/CD8⁺ T cell antitumor immunity induced by the combination therapy, we hypothesized that the activated CD8⁺ T cells might be responsible for the distant effect and evaluated the change in the number of CD8⁺ T cells in distant metastatic tumours.

Here, according to the suggestions of Reviewer 4 and the Editors, we performed a series of rescue experiments to evaluate the effect of depletion of CD8 or blockade of MHC-I on the antitumour abscopal effect (**Fig. 8a**). Single dose intraperitoneal administration of antibody drugs could achieve CD8⁺ T cell depletion in circulation, and MHC-I blockade on cDC1s infiltrating in TdLNs (**Fig. 8b**). The results showed that after treatment with IRE and Hydrogel Vaccine for subcutaneous tumor, CD8⁺ T cell depletion or MHC-I blockade significantly promoted the growth of distant subcutaneous metastases (**Fig. 8c**). The CD8⁺ T cell depletion or MHC-I blockade also promoted lung (**Fig. 8d, e**) and liver (**Fig. 8f, g**) metastases of Panc02 cells after combination therapy for the orthotopic Panc02 tumors. The CD11c-DTR transgenic mice are the classical method for specific depletion of CD11c⁺ cDC1s [5], and there is currently no effective antibody-based method for *in vivo* cDC1 depletion. Referring to previous studies, we prohibited the antigen-presenting function of cDC1s and its activation of CD8⁺ T cells using MHC-I blocking antibodies [6]. The results showed that the blockade of MHC-I also significantly promoted the growth of distant metastases. Due to the complexity of the immune system, it is challenging to exclude the possibility that other immune cells contribute to the abscopal effects induced by the combination therapy. However, combined with all the data we have presented, the results at least demonstrated the important role of cDC1s and CD8⁺ T cells in

inducing the abscopal effect in our study. Relevant revision can be found in **Results** section (*page 14, line 25-29; page 14, line 1-6*).

We thank the reviewer again for the valuable comment. We sincerely hope this explanation could clarify this issue.

References:

- [1] Chowdhury S., Castro S., Coker C., et al. Programmable bacteria induce durable tumor regression and systemic antitumor immunity. [J]. Nat Med, 2019, 25: 1057-1063.
- [2] Pan S., Guan J., Xianyu B., et al. A Nanotherapeutic Strategy to Reverse NK Cell Exhaustion. [J]. Adv Mater, 2023, undefined: e2211370.
- [3] Nishiga Y., Drainas A., Baron M., et al. Radiotherapy in combination with CD47 blockade elicits a macrophage-mediated abscopal effect. [J]. Nat Cancer, 2022, 3: 1351-1366.
- [4] Abe S., Nagata H., Crosby E., et al. Combination of ultrasound-based mechanical disruption of tumor with immune checkpoint blockade modifies tumor microenvironment and augments systemic antitumor immunity. [J]. J Immunother Cancer, 2022, 10, doi:10.1136/jitc-2021-003717
- [5] Jung S., Unutmaz D., Wong P., et al. In vivo depletion of CD11c⁺ dendritic cells abrogate priming of CD8⁺ T cells by exogenous cell-associated antigens. [J]. Immunity, 2002, 17: 211-220.
- [6] Baptista A., Gola A., Huang Y., et al. The Chemoattractant Receptor Ebi2 Drives Intranodal Naive CD4 T Cell Peripheralization to Promote Effective Adaptive Immunity. [J]. Immunity, 2019, 50: 1188-1201.e6.

Figure 8. *In vivo* depletion of CD8⁺ T cells or blockade of MHC-I impairs the abscopal effect induced by IRE and Hydrogel Vaccine combination therapy.

(a) Schematic diagram of the in vivo experiments. (b) Flow cytometry analysis demonstrated the efficiency of CD8 depletion and MHC-I blockade in mice bearing pancreatic tumors. (c) Comparison the volume of distant metastatic tumours across different groups (n=5 for each group). Statistical analyses were performed using Dunnett-t test. (d)

Representative images of H&E staining of lung metastases of Panc02 cells across different treatment groups. (e) In vivo depletion of CD8⁺ T cells or blockade of MHC-I significantly increases lung metastasis of Panc02 cells after the combination therapy of IRE and hydrogel vaccine. (f) Representative images of H&E staining of liver metastasis of Panc02 cells across different treatment groups. (g) In vivo depletion of CD8⁺ T cells or blockade of MHC-I significantly increases liver metastasis of Panc02 cells after the combination therapy of IRE and hydrogel vaccine. The error bar indicates the mean \pm s.e.m. in this figure.

[REDACTED]

REVIEWERS' COMMENTS

Reviewer #4 (Remarks to the Author):

In the revised manuscript and response letter, the authors have tried to address the comments of the reviewer with further clarification and additional data. The authors explained the material manufacturing mechanisms and managed to show that the release rates of CD40L at pH 6.8 and pH 7.4 are statistically different.

The authors are suggested to address these two remaining issues:

1. In response to comment 1, the authors confirmed the reviewer's comment that there are previous publications on biomaterials activating cDC1s which is also the strategy of the current work. The authors did not explain how/why the current study can be set apart from previous works. It is important to clarify what the current study contributes to the field.

2. In response to comment 12, the authors performed an MHC-I blocking study. However, this only shows that cross-presentation via MHC-I is crucial for the therapeutic effect of their materials but this does not prove the claim that their materials achieved the therapeutic outcomes by cDC1-mediated antigen cross-presentation. The cross-priming of cytotoxic T cells can also be mediated by other APCs. Therefore, please consider revising this claim in the manuscript.

Responses to Reviewers

We thank the reviewers for their valuable comments and support on this research work, which have guided the revision of this manuscript (**Manuscript ID: NCOMMS-22-11357B**). The concerns have been addressed as follows:

Reviewer #4 (Remarks to the Author):

In the revised manuscript and response letter, the authors have tried to address the comments of the reviewer with further clarification and additional data. The authors explained the material manufacturing mechanisms and managed to show that the release rates of CD40L at pH 6.8 and pH 7.4 are statistically different.

The authors are suggested to address these two remaining issues:

Comment 1: In response to comment 1, the authors confirmed the reviewer's comment that there are previous publications on biomaterials activating cDC1s which is also the strategy of the current work. The authors did not explain how/why the current study can be set apart from previous works. It is important to clarify what the current study contributes to the field.

Answer: Thanks very much for the insightful comment. We agree with the reviewer that studies have demonstrated the technical feasibility of cDC1 activation or biomaterial-based drug delivery in pancreatic cancer. However, we believe that cDC1 activation and drug delivery in pancreatic cancer should not be evaluated separately in the current study. Due to the complex biological features of pancreatic cancer, it has been challenging in previous studies to directly activate cDC1 and achieve therapeutic effect in pancreatic cancer. In this study, we inhibited pancreatic cancer in preclinical murine model by combining ablation and cDC1 activation with high efficiency. This new combination therapy and its therapeutic effect go beyond the technical feasibility. Compared to previous studies, this study also highlights the potential translational value of ablation-immunotherapy in the treatment of locally advanced pancreatic cancers.

Based on suggestions from the reviewer and editors, we have made revisions in the *Introduction* sections. We sincerely hope this explanation could clarify this issue.

Comment 2: In response to comment 12, the authors performed an MHC-I blocking study. However, this only shows that cross-presentation via MHC-I is crucial for the therapeutic effect of their materials but this does not prove the claim that their materials achieved the therapeutic outcomes by cDC1-mediated antigen cross-presentation. The cross-priming of cytotoxic T cells can also be mediated by other APCs. Therefore, please consider revising this claim in the manuscript.

Answer: Thanks very much for the valuable comment. Through the rescue experiments, we have demonstrated that CD8⁺ T cell activation is essential for the antitumor effect of hydrogel vaccines. However, targeted cDC1 depletion with antibodies or genetically engineered mouse models is still challenging due to technical limitations. The subsequent experiments demonstrated that MHC-I, a crucial molecule for antigen presentation on cDC1, was essential for vaccine-induced antitumor immunity. Since we have demonstrated that the vaccine promoted the recruitment and activation cDC1s, it can be inferred that cDC1 has a promoting effect on antitumor immunity of CD8⁺ T cells. In contrast, as mentioned by the reviewers, this does not completely exclude the possible activation of CD8⁺ T cells by other APCs. This is also a limitation of this study. We have revised the claims and added the limitations in the *Discussion* section according to the reviewer's suggestion. We thank the reviewer again for the valuable comment.